# Globally Approved EGFR Inhibitors: Insights into Their Syntheses, Target Kinases, Biological Activities, Receptor Interactions, and Metabolism

**DOI:** 10.3390/molecules26216677

**Published:** 2021-11-04

**Authors:** Mohammed A. S. Abourehab, Alaa M. Alqahtani, Bahaa G. M. Youssif, Ahmed M. Gouda

**Affiliations:** 1Department of Pharmaceutics, Faculty of Pharmacy, Umm Al-Qura University, Makkah 21955, Saudi Arabia; maabourehab@uqu.edu.sa; 2Department of Pharmaceutical Chemistry, Faculty of Pharmacy, Umm Al-Qura University, Makkah 21955, Saudi Arabia; 3Pharmaceutical Organic Chemistry Department, Faculty of Pharmacy, Assiut University, Assiut 71526, Egypt; bgyoussif@ju.edu.sa; 4Department of Medicinal Chemistry, Faculty of pharmacy, Beni-Suef University, Beni-Suef 62514, Egypt

**Keywords:** EGFR, kinase inhibitor, synthesis, anticancer, metabolism

## Abstract

Targeting the EGFR with small-molecule inhibitors is a confirmed valid strategy in cancer therapy. Since the FDA approval of the first EGFR-TKI, erlotinib, great efforts have been devoted to the discovery of new potent inhibitors. Until now, fourteen EGFR small-molecule inhibitors have been globally approved for the treatment of different types of cancers. Although these drugs showed high efficacy in cancer therapy, EGFR mutations have emerged as a big challenge for these drugs. In this review, we focus on the EGFR small-molecule inhibitors that have been approved for clinical uses in cancer therapy. These drugs are classified based on their chemical structures, target kinases, and pharmacological uses. The synthetic routes of these drugs are also discussed. The crystal structures of these drugs with their target kinases are also summarized and their bonding modes and interactions are visualized. Based on their binding interactions with the EGFR, these drugs are also classified into reversible and irreversible inhibitors. The cytotoxicity of these drugs against different types of cancer cell lines is also summarized. In addition, the proposed metabolic pathways and metabolites of the fourteen drugs are discussed, with a primary focus on the active and reactive metabolites. Taken together, this review highlights the syntheses, target kinases, crystal structures, binding interactions, cytotoxicity, and metabolism of the fourteen globally approved EGFR inhibitors. These data should greatly help in the design of new EGFR inhibitors.

## 1. Introduction

The epidermal growth factor receptor (EGFR), also named ErbB-1 and Her1, is a transmembrane protein. It is classified as a tyrosine kinase. The EGFR is also a member of the ErbB family which includes EGFR (ErbB-1) and three other members, HER2 (ErbB-2), HER3 (ErbB-3), and HER4 (ErbB-4) [1].

EGFR is composed of three subunits (Figure 1). The first is the extracellular EGF binding domain, the second is a transmembrane domain, and the third is a cytoplasmic domain [2]. The cytoplasmic domain consists of two subunits, the protein tyrosine kinase (PTK) domain and the C-terminal phosphorylation domain. 

The overexpression of the EGFR is associated with different types of cancers [3,4]. Accordingly, targeting the EGFR with small-molecule inhibitors represents a promising strategy in cancer therapy [5]. This area has attracted the attention of researchers during the last two decades, in which a large number of EGFR small-molecule inhibitors have been developed [6,7,8]. 

Following the binding of the epidermal growth factor (EGF) or the other EGF-like ligands such as transforming growth factor alpha (TGF-α) to EGFR, a downstream signaling cascade is initiated, as shown in Figure 1. The signaling cascades can be divided into two pathways [9]. The first axis mediates the effects of EGFR on cell cycle progression and proliferation. This axis includes the KRAS-RAF-mitogen-activated protein kinase (MAPK) axis. The second axis mediates the anti-apoptosis signals and is mediated by the phosphatidylinositol 3-kinase (PI3K) pathway. It was also reported that STAT proteins can mediate intracellular EGFR signaling by direct/indirect activation mechanism [10].

The kinase domain in EGFR is a part of the cytoplasmic region of the kinase. The transfer of a phosphate group to the kinases is mediated by this domain [11]. The kinase domain (Figure 2) consists of two lobes, designated as N- and C-lobes [12]. The ATP binding pocket is located between these two lobes [13]. Besides the binding of the adenine base of ATP to this pocket, it can also accommodate competitive-type EGFR-TKIs [14]. The crystal structure of the wild-type EGFR (pdb: 3VJO) [15] bound to adenylylimidodiphosphate (AMPPNP), a non-hydrolysable analogue of ATP, is visualized in Figure 2. AMPPNP exhibits two conventional hydrogen bonds with Met793 and Gln791. 

Discovery Studio Visualizer [16] was used to generate the binding modes/interactions figures of all the EGFR inhibitors discussed in this review. 

Currently, more than 100 crystal structures of EGFR bound to small-molecule inhibitors are available in the protein data bank (https://www.rcsb.org/) (accessed on 8 September 2021). These crystal structures provide a huge amount of information that enables researchers in this field to understand the structure and function of the EGFR. This must enhance the discovery and development of rationally designed EGFR tyrosine kinase inhibitors [17,18,19]. 

Since the approval of the first rationally designed kinase inhibitor, imatinib (Figure 3), in 2001, the design of new kinase inhibitors has attracted a high amount of attention [20]. Two years later, the first EGFR-TKI, gefitinib, was approved for the treatment of non-small cell lung cancer (NSCLC) [21]. On 18 November 2004, the second EGFR-TKI, erlotinib, also received FDA approval for the treatment of NSCLC [22]. 

On 13 March 2007, the FDA approved the third EGFR-TKI, lapatinib, in combination with capecitabine for the treatment of breast cancer [23]. However, in the second decade of the 21st century, eleven EGFR-TKIs were approved for the treatment of different types of cancers. These drugs can be divided into two groups. The first one includes seven EGFR-TKIs approved by the FDA, including afatinib, brigatinib, dacomitinib, neratinib, osimertinib, pyrotinib, and vandetanib. The second group includes four drugs approved for the treatment of cancers outside the USA. This group includes three EGFR-TKIs, icotinib, almonertinib, and simotinib, that were approved in China between 2011 and 2020. In addition, olmutinib was approved in South Korea in 2016. 

In the current review, the primary focus is on the small-molecule inhibitors of EGFR which are approved for clinical use as anticancer agents. The classifications, approval history, syntheses, kinase inhibitory activities, receptor interactions, cytotoxicity, and the metabolic pathways of these approved drugs are discussed.

### 1.1. Classification of the Approved EGFR-TKIs

#### 1.1.1. Chemical Classification

Based on their chemical structures, the approved EGFR-TKIs can be classified into three subclasses. The first subclass is the aminopyrimidine derivatives which include almonertinib, brigatinib, and osimertinib. In this study, olmutinib is considered as a fused aminopyrimidine derivative. The second group has a quinoline-4,6-diamine nucleus and includes two derivatives, neratinib and pyrotinib. The third group is the quinazoline-4,6-diamine-based derivatives which include afatinib, dacomitinib, erlotinib, icotinib, lapatinib, simotinib, and vandetanib, as shown in Figure 4. 

#### 1.1.2. Classification Based on the Types of Interaction with EGFR

Based on the type of inhibition of the EGFR kinase activity, the approved EGFR-TKIs can also be classified as reversible and irreversible inhibitors, as shown in Figure 5. For the first type, the inhibitors competitively bind to the ATP binding site in the EGFR through noncovalent interactions involving electrostatic, hydrogen-bonding, and hydrophobic interactions [24].

In the second type, the EGFR inhibitors form a covalent bond with the cysteine residue in the EGFR. The structure of these inhibitors is characterized by the presence of an electrophilic side chain that acts as a Michael acceptor which chemically reacts with the cysteine thiol group to form a covalent adduct [25].

#### 1.1.3. Classification Based on the Clinical Use

The EGFR inhibitors can also be classified into three groups based on the types of cancers for which they are approved [26]. The first group includes drugs which are approved for the treatment of NSCLC such as afatinib, almonertinib, brigatinib, dacomitinib, erlotinib, gefitinib, icotinib, olmutinib, and osimertinib. The second type includes drugs approved for the treatment of breast cancer such as lapatinib and neratinib, and pyrotinib. On the other hand, the third group includes drugs approved for other types of cancers such as thyroid cancer (vandetanib), pancreatic cancer (erlotinib), and solid cancers (simotinib).

#### 1.1.4. Classification Based on the Target Kinases

The approved EGFR-TKIs may also be classified according to their target kinases into (1) selective EGFR inhibitors which target EGFR with high selectivity such as gefitinib; (2) dual EGFR inhibitors such as lapatinib which can target EGFR and ErbB-2; and (3) multi-kinase inhibitors such as brigatinib, pyrotinib, and vandetanib. These drugs have broad-spectrum activity against multiple kinases other than the EGFR [27]. 

#### 1.1.5. Classification into the First, Second, and Third Generation

Among the approved EGFR-TKIs, gefitinib, erlotinib, lapatinib, and icotinib are classified as first-generation EGFR inhibitors, as shown in Figure 6. The drugs in this class bind reversibly to the PTK domain of the EGFR, which leads to the inhibition of the binding of ATP to the EGFR and consequently to the inhibition of EGFR activation and cellular proliferation [28]. The chemical structures of the four drugs consist of a basic quinazoline nucleus attached to a substituted aniline moiety, as shown in Figure 3. 

On the other hand, afatinib, neratinib, and dacomitinib are classified as second-generation EGFR-TKIs. These three drugs contain a Michael acceptor site which allow them to bind covalently to the EGFR, leading to the irreversible inhibition of the kinase activity, which provides an advantage over first-generation EGFR-TKIs [29]. The chemical structures of these drugs consist of quinazoline or a quinoline nucleus which bears a crotonamide side chain (Michael acceptor site) substituted at the terminal carbon by a *tert*-amino group. 

The third-generation EGFR inhibitors include almonertinib, olmutinib, and osimertinib [30]. The chemical structures of these drugs include a pyrimidine nucleus attached to a substituted anilino or phenoxy moiety. These moieties bear an acrylamide group that, as noted above, can form a covalent bond with the cysteine residue in the EGFR. 

On the other hand, brigatinib, vandetanib, and pyrotinib are classified as multi-kinase inhibitors due to their inhibitory activities against kinases other than the EGFR [31].

The emergence of EGFR T790M and C797S mutation has led to the rapid development of resistance to the first- second- and third-generation EGFR-TKIs [30]. The C797S mutation also deprives the irreversible inhibitors of the EGFR of the ability to bind covalently to the kinase. In addition, the emergence of EGFR double and triple (del19/L858R + C797S ± T790M) mutations challenges the therapeutic effectiveness of EGFR-TKIs. These problems underscore the ongoing need to develop new and potent EGFR-TKIs.

Recently, several fourth-generation allosteric EGFR inhibitors that bind to a site in the EGFR other than the PTK domain were reported [32]. However, although these inhibitors are ineffective against NSCLC with a mutated EGFR, they display synergistic anticancer effects when combined with an EGFR inhibitor such as osimertinib or the monoclonal antibody, cetuximab.

Currently, several fourth-generation EGFR-TKIs are being evaluated in regard to their efficacy against cancers carrying double or triple EGFR mutations [33,34,35]. One of these compounds, BBT-176, is in Phase I/II trials in NSCLC patients with advanced lung cancer [35], but these inhibitors are not yet approved for clinical use.

### 1.2. EGFR Inhibitors Approved for Cancer Treatments

#### 1.2.1. Afatinib

##### Approval History

Afatinib (Figure 7) is a second-generation quinazoline derivative which binds covalently to the intracellular PTK domain of EGFR. On 12 July 2013, Afatinib was approved by the FDA for use in the treatment of patients with metastatic NSCLC with a mutation in the EGFR (EGFR exon 19 deletions or exon 21 (L858R) substitution mutations) [36]. In addition, in 2016, afatinib was also approved for the treatment of squamous cell carcinoma of the lung [37]. Afatinib indication was broadened to metastatic NSCLC, whose tumors have a non-resistant EGFR [38]. 

##### Synthesis

The synthesis of afatinib (Figure 1) from the commercially available 2-amino-4-fluorobenzoic acid 1 was reported by Kovacevic et al. [39]. The iodination of compound 1 using sodium periodate and potassium iodide afforded compound **2** which underwent intramolecular cyclization with formamide acetate to give the dihydroquinazoline **3.**

The etherification of compound **3** with tetrahydrofuran-3-ol **4** proceeded through a dehydrohalogenation reaction and gave compound **5**, as shown in Figure 1. The chlorination of compound **5** with thionyl chloride yielded 4-chloro-6-iodoquinazoline **6**, which was reacted with 3-chloro-4-fluoroanilin **7** to give compound **8**. The reaction of **8** with (*E*)-4-(dimethylamino)but-3-enamide **9** afforded afatinib [39]. 

##### Target Kinases

Afatinib inhibits the kinase activity of the wild-type EGFR at an IC_50_ value of 31 nM [40]. In addition, it shows inhibitory activity against several EGFR mutations. The IC_50_ values of afatinib against wild-type and EGFR mutations are presented in Figure 8. 

Afatinib showed higher inhibitory activities against the both a wild-type and mutant EGFR compared to erlotinib, as shown in Figure 8. It showed an IC_50_ value of 0.2 nM against both the EGFR (Exon 19del), and EGFR (L858R). On the other hand, afatinib exhibited lower inhibitory potency against the EGFR (Exon 19del + T790M), and EGFR (L858R + T790M) [40]. In another study, afatinib exhibited inhibitory activity against the HER2 kinase and ErbB-4 kinase at EC_50_ values of 14 and 1 nM, respectively [41].

##### Crystal Structures and Binding Interactions

Afatinib was found as a bound ligand in three crystal structures in the protein data bank. These crystals include the crystal structure of afatinib bound to the EGFR kinase (pdb: 4G5J) [41], EGFR kinase T790M (pdb: 4G5P) [41], and human brachyury G177D variant (pdb: 6ZU8). 

The binding interactions of afatinib into EGFR kinase (pdb: 4G5J) are visualized in Figure 9. Afatinib displayed one conventional hydrogen bond (3.3 Å) with Met793 [41]. In addition, two carbon hydrogen bonds were formed with Lys728 and Gln791. Afatinib also showed one halogen interaction with Glu762 and multiple hydrophobic interactions with the EGFR. 

The binding interactions of afatinib into the EGFR kinase T790M (pdb: 4G5P) are also visualized in Figure 10. These interactions were similar to those formed by afatinib in the wild-type EGFR (Figure 9). In the active site of EGFR kinase T790M, afatinib exhibited one hydrogen binding interaction with Met793, one carbon hydrogen bond with Gln791, halogen interaction with Glu762, and multiple hydrophobic interactions (Figure 10). 

However, besides these reversible binding interactions, afatinib was also reported to irreversibly inhibit the ErbB receptor family members. This inhibition depends on the ability of afatinib to form a covalent bond with the cysteine residues in EGFR, HER2, and HER4, which results in the inhibition of the tyrosine kinase activity [41,42]. The mechanism of interaction depends on the presence of the Michael acceptor site which reacts with the thiol group in Cys797 in the EGFR. The formation of this covalent bond leads to the irreversible inhibition of the autophosphorylation of the ErbB family [41]. The mechanism of the irreversible inhibition of the EGFR by afatinib is illustrated in Figure 11. 

The covalent interaction of afatinib with the ErbB receptor family is not restricted to the wild-type EGFR. Afatinib binds to and inhibits the EGFR with exon 19 deletion mutation and exon 21 L858R mutation. The covalent interaction leads to the irreversible inhibition of the kinase activity, which provides an advantage over other non-covalent competitive inhibitors of the EGFR such as erlotinib and gefitinib [43,44,45]. 

##### Biological Activity

Afatinib binds to the EGFR, HER2, and ErbB4 and inhibits tyrosine kinase activity, leading to the inhibition of the intracellular signaling pathways and the inhibition of cell growth [46]. Due to these advantages over first-generation EGFR inhibitors, afatinib also was investigated against cancer cells exhibiting abnormalities of the ErbB network such as NSCLC and breast cancer [47]. In addition, the results of the in vitro evaluation of the anticancer activity of afatinib against a series of cancer cell lines expressing EGFR mutations revealed superior activity compared to the first-generation EGFR inhibitor, erlotinib [40]. 

Evaluation of the anticancer activity of afatinib against several lung cancer cell lines, including PC-9 (exon 19del), H3255 (L858R), PC-9ER (exon 19del + T790M), H1975 (L858R + T790M), and BID007 (A763-Y764insFQEA), revealed IC_50_ values in the range of 0.3-165 nM compared to osimertinib (IC_50_ = 4–40 nM). This study also indicated higher anticancer activity for afatinib against PC-9 (exon 19del), H3255 (L858R), and BID007 (A763-Y764insFQEA) cell lines compared to osimertinib [40].

##### Metabolism 

Several studies were performed to prepare radiolabeled derivatives of afatinib, which can be used to investigate the metabolic profile of afatinib. Slobbe et al. [48] reported a synthetic method for [^18^F] afatinib as a new TKI-PET tracer. The [^18^F] afatinib tracer displayed good stability in vivo and was used to study the metabolism and biodistribution of afatinib [48]. The results of this study revealed a moderate rate of metabolism and rapid tumor accumulation compared to background tissues. 

On the other hand, Stopfer et al. studied the metabolism of afatinib in healthy male volunteers using the [^14^C]-radiolabeled derivative [49]. The study revealed that the cytochrome P-450-mediated oxidative metabolism was of negligible importance. The main route of excretion of was via feces, where 85.4% of [^14^C] radioactivity was excreted in feces. These results indicated that afatinib undergoes minimal metabolism. 

In addition, the dimethylamino *N*-oxide metabolite was also detected among minor metabolites. The main metabolic pathway of afatinib was an adduct (Figure 12) formed by the addition of electron-rich groups in glutathione (GSH), cysteine, or plasm protein to the Michael acceptor site in afatinib [49]. 

A recent study performed by Wind et al. [50] also revealed a minimal metabolism of afatinib. Excretion in feces was indicated as the main route of excretion, while 5% of the drug was excreted in the urine. The presence of the Michael acceptor site in afatinib not only affects the pharmacodynamics of the drug but has also a clear effect on its pharmacokinetics. Afatinib showed an effective half-life of 37 h, while an estimated terminal half-life of 344 h was observed in patients who received afatinib for more than 6 months. This could be attributed to the covalent adduct formed between afatinib and proteins. The slow decomposition of this adduct could lead to the prolongation of the elimination phase [42]. 

#### 1.2.2. Almonertinib

##### Approval History

Almonertinib (Figure 13) is classified as a third-generation EGFR-TKI. In 2020, almonertinib was approved for use in the treatment of patients with advanced EGFR T790M mutation-positive NSCLC [51]. 

##### Synthesis

The synthesis of almonertinib (Figure 1) was achieved from the commercially available indole [52]. Compound **12** was prepared by the alkylation of the indole **10** using cyclopropyl boronic acid. The coupling of compound **12** with 2,4-dichloropyrimidine **13** afforded compound **14**. 

The reaction of compound **14** with 4-fluoro-2-methoxy-5-nitroaniline **15** yielded compound **16**, which underwent another coupling reaction with *N*^1^*,N*^1^*,N*^2^-trimethylethane-1,2-diamine **17** to give compound **18**, as shown in Figure 2. The reduction of the nitro group in compound **18** followed by the acylation with 3-chloropropionyl chloride and dehydrohalogenation with KOH afforded almonertinib as a free base. 

##### Target Kinases

Considering the definition of me-too drugs [53], almonertinib could be considered as a me-too drug of osimertinib, the first-in-class compound. The only structural difference is that the methyl group at the indole nitrogen in osimertinib was replaced by a cyclopropyl moiety in almonertinib. Accordingly, the two drugs have similar target kinases and clinical uses. The enzymatic activity of almonertinib (Figure 14) revealed inhibitory activity against the wild-type EGFR at an IC_50_ value of 3.39 nM [54]. However, it showed more than ten times higher inhibitory activities against the mutant types, EGFR T790M, T790M/L858R, and T790M/Del19 (IC_50_ = 0.21–0.37 nM), compared to the wild type. 

##### Crystal Structures and Binding Interactions

The crystal structure of almonertinib bound to any of its target kinases has not yet been reported.

##### Biological Activity

Almonertinib also inhibited the phosphorylation of the wild-type EGFR in a cell-based assay at an IC_50_ value of 596.6 nM in the A431 (wild-type EGFR) cell line [30]. However, higher inhibitory activities were observed in HCC827 (EGFR del19), PC9 (EGFR del19), and NCI-H1975 (EGFR L858R/T790M) cancer cell lines harboring an EGFR mutation (IC_50_ = 3.3–4.1 nM). 

Almonertinib was also investigated against other targets which could contribute to the anticancer activity. Wu et al. have reported the ability of almonertinib to reverse multi-drug resistance (MDR) in cancer cells overexpressing ABCB1 transporter [55].

##### Metabolism

The metabolism of [^14^C] almonertinib was investigated by Zhou et al. [56] in healthy male participants. The results revealed 26 metabolites that were identified in human blood, urine, and feces. Representative pathways to these metabolites are presented in Figure 15.

The study revealed that the elimination of [^14^C] HS-10296 takes place via feces after secretion in the bile [56]. The metabolic pathways of almonertinib includes demethylation, which gives M511a,b and oxidative dealkylation of the 2′-(*N*,*N*-dimethylamino)ethyl group that gives M2, which underwent oxidation or coupling with cysteine/acetylcysteine to give M575 and M617, respectively. Among the reported metabolite of almonertinib, HAS-719 (M511b) displayed lower inhibitory activities against EGFR mutations compared to the parent drug, almonertinib [30]. 

#### 1.2.3. Brigatinib

##### Approval History

Brigatinib (Figure 16) is a phosphorous derivative with multi-kinase inhibitory activities. It was approved by the FDA on 28 April 2017 for use in the treatment of patients with ALK-positive NSCLC [57]. 

##### Synthesis

The synthesis of brigatinib (Figure 3) was reported by Huang et al. [58]. Brigatinib was obtained from the coupling of compounds **23** and **26**. The intermediate **22** was prepared from the reaction of the substituted nitrobenzene **20** with the piperazine derivative **21**. 

Compound **23** was obtained via the hydrogenation of the nitro group in **22** using hydrogen in the presence of palladium/carbon as a catalyst. On the other hand, the Pd-catalyzed cross-coupling of **23** and **26** in DMF afforded brigatinib, as shown in Figure 3. 

##### Target Kinases

Brigatinib is a multikinase inhibitor that targets ALK, ROS1, FLT3, and mutant EGFR [59]. The inhibitory activity of brigatinib against selected kinases is illustrated in Figure 17. These IC_50_ values indicated that brigatinib has higher inhibitory activity against ALK than the wild-type EGFR. 

##### Crystal Structures and Binding Interactions 

Brigatinib exists in two crystal structures in the protein data bank. The first crystal structure in a complex of brigatinib with the EGFR C797S mutation (pdb: 7AEM) [60]. The second crystal structure (pdb: 6MX8) consists of brigatinib in complex the anaplastic lymphoma kinase (ALK) [58]. 

Visualization of the binding modes and interactions of brigatinib into the EGFR C797S mutation revealed one conventional hydrogen bond with Met793 and two carbon hydrogen bonds with Gln791 and Pro794, as shown in Figure 18. In addition, multiple hydrophobic interactions could also be observed with the EGFR, also shown in Figure 18. 

Brigatinib also displayed one conventional hydrogen bond with Met1199 in ALK (pdb: 6MX8), as shown in Figure 19. In addition, it showed three carbon hydrogen bonds with Leu1122, Glu1197, and Ala1200. One water molecule was also involved in one water hydrogen bond with brigatinib. 

##### Biological Activity 

The anticancer activity of brigatinib against lung cancer was mediated by the inhibition of the EGFR, ALK, FLT3, and other kinases [61]. The cell viability assay of brigatinib revealed IC_50_ values of 29 nM and 3194 nM against Karpas-299 (ALK+) and U937 (ALK-) cell lines [58]. The advantage of brigatinib over the remaining EGFR inhibitors is that it acts as a dual inhibitor for both ALK and the EGFR [61]. Brigatinib was also effective against the mutant variant of the EGFR and ALK, which are resistant to common types of inhibitors [61]. In addition, brigatinib has variable degrees of inhibitory activities against other oncogenic kinases which contribute to the proliferation of cancer cells. After the accelerated approval in 2017, brigatinib was used for the treatment of NSCLC ALK+ patients [57]. 

##### Metabolism

In healthy subjects, radiolabeled brigatinib underwent partial metabolism through two different metabolic pathways, including *N*-demethylation and cysteine conjugation. Of the total dose (180 mg) of brigatinib given to the healthy subjects, 92% remained unchanged, while 3.5% was metabolized to the primary metabolite (AP26123) [61].

The in vitro metabolism of brigatinib using rat liver microsomes was assessed by Kadi et al. [62]. The study revealed that brigatinib is metabolized through three metabolic pathways, including the *N*-dealkylation, α hydroxylation and α-oxidation pathways, as shown in Figure 20. Potassium cyanide was used to trap the reactive iminium intermediates, which were proposed to be formed from the metabolism of the piperidine ring.

#### 1.2.4. Dacomitinib

##### Approval History

On 27 September 2018, dacomitinib (Figure 21) was approved by the FDA for use in the treatment of metastatic NSCLC with EGFR exon 19 deletion or exon 21 L858R substitution mutations [63]. Dacomitinib is classified as an irreversible inhibitor of HER1 (EGFR), HER2, and HER4.

##### Synthesis 

Dacomitinib could be achieved in 58% overall yield using a three-step synthesis [64], as shown in Figure 4. The reduction of the nitro group in compound **27** afforded compound **28**. The acylation of **28** with (E)-4-(piperidin-1-yl)but-2-enoic acid **29** was catalyzed by 1-propanephosphonic acid cyclic (T3P) and 2,6-lutidine to give compound **30**. Upon the reaction of compound **30** with the substituted aniline **31**, Dimroth rearrangement took place, giving dacomitinib. 

##### Target Kinases 

The kinase inhibitory activity of dacomitinib was evaluated against the wild type of the HER family [65]. The results revealed IC_50_ values of 6.0, 45.7, and 73.7 nM against EGFR, HER2, and HER4, respectively, as shown in Figure 22.

##### Crystal Structures and Binding Interactions 

Dacomitinib exists in two crystal structures in the protein data bank. These crystals include dacomitinib in complex with the wild-type EGFR kinase domain (pdb: 4I23) and T790M EGFR kinase domain (pdb: 4I24) [66].

The binding mode and interactions of dacomitinib into EGFR T790M kinase domain (pdb: 4I24) are visualized in Figure 23. Dacomitinib exhibited one covalent bond with Cys797, one conventional hydrogen bond with Met793, one halogen interaction with Leu788, one pi-donor hydrogen bond with Thr854, and three carbon hydrogen bonds with Gln791 and Pro794. 

##### Biological Activity 

Evaluation of the growth inhibitory activity of dacomitinib against a panel of 27 head and neck squamous cancer cell lines was performed by Ather et al. [67]. The results revealed a reduction in the growth of the cancer cells by at least 50% in 17 of the tested cell lines compared to erlotinib, which showed similar growth inhibition in 7 cell lines. 

##### Metabolism

Attwa et al. investigated the phase I metabolism of dacomitinib [68]. In this study, potassium cyanide was used to capture reactive metabolites such as aldehydes and iminium intermediates. The study revealed the formation of four metabolites, where the hydroxylation of the piperidine ring was the major pathway, as shown in Figure 24. 

#### 1.2.5. Erlotinib

##### Approval History

Erlotinib (Figure 25) was approved by the FDA on 18 November 2004 for use in the treatment of the locally advanced or metastatic NSCLC [22]. On 2 November 2005, it was also approved in combination with gemcitabine as a first-line treatment for patients with pancreatic cancer [69].

##### Synthesis 

Several synthetic routes were reported that described the synthesis of erlotinib using diverse reagents and reaction conditions [70,71,72]. Among these methods, the method reported by Barghi et al. [70] described an eight-step synthesis of erlotinib, as shown in Figure 5. 

In the first step, the hydroxyl and carboxylic acid groups in 3,4-dihydroxybenzoic acid **32** underwent alkylation using 1-chloro-2-methoxyethane to give compound **33**. Hydrolysis of the ester group in **32** via alcoholic potassium hydroxide afforded the carboxylic acid derivative **34**, which underwent esterification with ethanol to give the ethyl ester **35**. The nitration of **35** followed by the subsequent reduction of the nitro group yielded compound **37**, which underwent intramolecular cyclization using a mixture of ammonium formate/formamide to give the dihydroquinazoline **38**. The treatment of compound **38** with oxalyl chloride afforded compound **39**, which was reacted with 3-ethynylaniline **40** to give erlotinib, as shown in Figure 5.

In addition, Asgari et al. [73] also reported another improved and economical method for erlotinib synthesis. In this method, a one-pot reaction of 2-aminobenzonitrile intermediate and 3-ethynylaniline was used to prepare the 4-anilinoquinazoline product. 

##### Target Kinases 

The kinase inhibitory activity of erlotinib was evaluated against the wild-type and mutated EGFR [40]. The results revealed higher inhibitory activity against the EGFR with exon 19del and L858R compared to the wild-type EGFR, as shown in Figure 26. 

Other studies were also performed to evaluate the kinase inhibitory activities of erlotinib [74,75]. The differences in the IC_50_ values could be attributed to the difference in the assay conditions. In addition, evaluation of the enzymatic activity of erlotinib was also performed against kinases other than the EGFR [75,76]. The results revealed inhibitory activity against JAK2 (V617F), which revealed an IC_50_ value of 4 μM of erlotinib [76].

##### Crystal Structures and Binding Interactions 

Erlotinib was found as a bound ligand in three crystal structures, including the EGFR tyrosine kinase domain (pdb: 1M17) [12], inactive EGFR tyrosine kinase domain (pdb: 4HJO) [77], and human cytochrome P450 1A1 (pdb: 6DWN) [78]. 

The binding modes/interactions of erlotinib in the first wild-type EGFR are illustrated in Figure 27. In addition to the multiple hydrophobic interactions, erlotinib displayed one conventional hydrogen bond with Met769, and two carbon hydrogen bonds with Gln767 and Thr830. One water molecule was found in the active site and mediated a hydrogen bond between erlotinib and the EGFR [12]. 

The binding modes/interactions of erlotinib into the inactive EGFR (pdb: 4HJO) are illustrated in Figure 28. One conventional hydrogen bond with Met769 and three carbon hydrogen bonds with Leu764, Gln767, and Thr830 could be observed in the interactions of erlotinib. In addition, two water molecules also contributed to the binding interactions of erlotinib through water hydrogen bonds [77]. 

##### Biological Activity 

Hirano et al. [40] evaluated the anticancer activity of erlotinib against several lung cancer cell lines harboring different types of EGFR mutations. Erlotinib showed IC_50_ values in the range of 7–1185 nM against PC-9 (exon 19del), H3255 (L858R), H1975 (L858R + T790M), and BID007 (A763-Y764insFQEA) lung cancer cell lines. On the other hand, erlotinib inhibited the proliferation of PC-9ER (exon 19del + T790M) at an IC_50_ value > 10 μM.

##### Metabolism 

The metabolism of erlotinib in healthy male volunteers was investigated by Ling et al. [79]. The study was performed using a single oral dose of [^14^C] erlotinib hydrochloride. The study revealed that about 83% of the dose (100 mg) was excreted in feces compared to 8.1% in urine. The study also revealed that erlotinib undergoes extensive metabolism (only 2% recovered unchanged). Based on the results of this study, the metabolic pathways of erlotinib could be classified into three major biotransformations. The first is the *O*-demethylation followed by oxidation of the resulting alcohol to give the carboxylic acid derivative M11 (29.4%). The second pathway is the oxidation of the ethynyl group to carboxylic acid derivative M6 (21.0%). The last metabolic pathway is the hydroxylation of the phenyl ring (9.6%), which affords a phenolic metabolite that underwent sulfate/glucuronic acid (GU) conjugation, as shown in Figure 29. Among these metabolites, M14, a pharmacologically active metabolite, was identified [79]. 

#### 1.2.6. Gefitinib

##### Approval History

Gefitinib (Figure 30) is a TKI indicated for use in the treatment of patients with metastatic NSCLC whose tumors have specific EGFR mutations [21]. It was first approved on 5 May 2003 for the treatment of cancer [21]. On 13 July 2015, it was approved for metastatic EGFR mutation-positive NSCLC [80].

##### Synthesis

The literature review revealed several synthetic pathways that can be used to prepare gefitinib [71,81]. Among these, a four-step method reported by Suh and co-workers [82] is illustrated in Figure 6.

The synthesis of gefitinib started with 7-methoxy-4-oxo-3,4-dihydroquinazolin-6-yl acetate **41** which underwent chlorination by phosphorus oxychloride to give **42**, as shown in Figure 6. The produced 4-chloroquinazoline **42** was reacted with 3-chloro-4-fluoroaniline **43** in isopropanol alcohol (*i*-ProOH) to give **44**. Hydrolysis of the ester group in **44** afforded quinazolin-6-ol **45**, which gave gefitinib on the reaction with 4-(3-chloropropyl)morpholine **46**.

##### Target Kinases 

The kinase inhibitory activity of gefitinib and eight of the approved kinase inhibitors was evaluated by Kitagawa et al. [75]. The kinase inhibitory activities were measured against 310 of the human kinases using an activity-based kinase assay. The assay revealed that gefitinib specifically inhibits the EGFR and its mutants. The results expressed as IC_50_ values are presented in Figure 31. 

The kinase inhibitory activity of gefitinib was also evaluated by Brehmer et al. [83]. Besides the inhibition of the EGFR, the results also revealed the ability of gefitinib to inhibit the serine/threonine kinases RICK and GAK (IC_50_ = 50 and 90 nmol/L, respectively). These results suggested that gefitinib could have alternative cellular modes of action. In addition, the cellular IC_50_ values of gefitinib against various EGFR mutants were also determined by Kancha et al. [74]. 

##### Crystal Structures and Binding Interactions

Gefitinib exist as a co-crystallized ligand in eight crystal structures in PDB. These crystals include gefitinib in complex with the EGFR kinase domain (pdb: 2ITY and 4WKQ) [84]. They also include the crystal structure of gefitinib bound to the mutated EGFR kinase (pdb: 2ITO, 2ITZ, 3UG2, 4I22, 5Y80, and 5Y7Z) [15,66,84,85].

The binding mode and interactions of gefitinib bound to the EGFR kinase domain (pdb: 2ITY) are visualized in Figure 32. Gefitinib exhibited one hydrogen bond with Met793, and several carbon hydrogen bonds with Leu718, Gln791, Pro794, and Gly796. 

The binding mod and interactions of gefitinib bound to the EGFR kinase domain L858R mutation (pdb: 2ITZ) are visualized in Figure 33. Gefitinib also showed one hydrogen bond with Met793, and several carbon hydrogen bonds with Lys745, Gln791, Pro794, Gly796, and Asp800. 

##### Biological Activity 

The in vitro activities of gefitinib against sensitive and resistant cancer cell lines were assessed in several reports [86,87]. Ono et al. [88] investigated the effect of gefitinib on the cell proliferation of nine NSCLC cell lines, including PC9, A549, H522, H322, H358, EBC-1, H157, QG56, and LK2 cell lines using MTS assay and colony formation assays. The results revealed IC_50_ values in the range of 4–42 μM (MTS assay). The results of this study revealed the association of growth inhibition on the activation of Akt and ERK1/2 in response to EGFR signaling. 

##### Metabolism 

Several studies were performed to investigate the metabolism of gefitinib by human cytochrome P450 enzymes [89,90,91]. Mckillop et al. studied the CYP450-based metabolism of [^14^C] gefitinib using human liver microsomes [91]. The aim of this study was focused on the M523595, an *O*-demethylated metabolite of gefitinib. The study revealed that gefitinib underwent an extensive hepatic metabolism by CYP3A4. the *O*-demethylation, oxidative defluorination, and oxidation of gefitinib were among the metabolic pathways identified in this study, as shown in Figure 34. 

Wang et al. investigated the metabolism of gefitinib in NSCLC patients [92]. Eighteen metabolites were tentatively identified in human plasma. Among these metabolites, M7 (Figure 35) was proposed to be formed through the removal of the 3-chloro-4-fluoroaniline moiety. In addition, dechlorination, and defluorination metabolism were proposed to give the M1-u, and M5 metabolites. The conjugation of M1-u with taurine was also proposed to produce the M1 metabolite. The conjugation of the phenolic metabolite M5 with sulfate gave the sulfate conjugate M4, while the demethylation of gefitinib was proposed to produce the phenolic M12 metabolite, which could undergo glucuronidation, yielding M8.

#### 1.2.7. Icotinib

##### Approval History

*Icotinib* (Figure 36) is a highly selective EGFR-TKI that was approved in China in June 2018 for use in the treatment of NSCLC [93]. 

##### Synthesis 

The synthesis of icotinib [93] was achieved from the starting materials **47** and **49** according to Figure 7. In the first step, compound **48** was obtained from the reaction of **47** with tosyl chloride. 

The reaction of compound **48** and ethyl 3,4-dihydroxybenzoate **49** in DMF afforded compound **50**, as shown in Figure 7. Compound **50** underwent nitration followed by the reduction of the nitro group to give compound **52**. The quinazolinone **53** was obtained by refluxing compound **52** with formamide. The action of compounds **53** with phosphorus oxychloride afforded 4-chloroquinazoline **54**, which undergoes a coupling reaction with 3-ethynylaniline **55** followed by hydrochloric acid to give icotinib hydrochloride salt.

Moreover, other studies describing the synthesis of icotinib and its derivatives were also reported [94,95].

##### Target Kinases 

The kinase inhibitory activity of icotinib was evaluated against 88 kinases by Tan et al. [96]. The results revealed variable inhibitory activities against the tested kinases. Among these kinases, the inhibitory activity against the human wild-type and mutant EGFR were meaningful, as shown in Figure 37. At 0.5 μM concentration, icotinib inhibited the activities of five EGFR kinases by 61–99%. The results also revealed that icotinib inhibits the activity of the EGFR at an IC_50_ value of 5 nM. 

##### Crystal Structures and Binding Interactions 

Until now, icotinib was not reported in crystal structures in the protein data bank. In addition, very limited studies were performed to evaluate the binding mode/interactions of icotinib or its derivatives [97,98]. However, these studies focused on targets other than the EGFR.

##### Biological Activity 

Tan et al. [96] also evaluated the anticancer activity of icotinib against seven cancer cell lines. To a large extent, the results (IC_50_ values) were dependent on the level of EGFR expression in the tested cancer cell lines. Among the seven cancer cell lines used in this study, A431 and BGC-823 were the most sensitive to icotinib (IC_50_ values of 1 and 4.06 M, respectively). In addition, icotinib inhibited the proliferation of A549, H460, and KB cell lines at IC_50_ values in the range of 12.16–40.71 μM. 

The anticancer activity of icotinib was also assessed in A549 lung cancer cells [99]. The results revealed an IC_50_ value of 8.8 µM for icotinib compared to 10.2 µM for gefitinib. 

##### Metabolism 

Chen et al. [100] reported that the metabolism of icotinib mainly depends on CYP3A4. In addition, Zhang et al. [101] also investigated the in vitro metabolism of icotinib using human liver microsomes and recombinant CYP isozymes. The results of the study revealed six major metabolites, and CYP3A4 was the most predominant enzyme in this metabolism (~87%). The proposed metabolites are illustrated in Figure 38. 

#### 1.2.8. Lapatinib

##### Approval History

Lapatinib (Figure 39) is a dual EGFR/ErbB-2 kinase inhibitor that was first approved by the FDA in 2007, in combination with capecitabine for advanced or metastatic breast cancer [23]. In February 2010, lapatinib received FDA approval as a first-line combination treatment for metastatic breast cancer [102].

##### Synthesis 

Erickson et al. [103] reported a new synthetic route of lapatinib via a regioselective alkylation reaction, as shown in Figure 8. In this synthesis, lapatinib was prepared from 2-amino-5-bromobenzoic acid **56**, which underwent cyclization with formamide to give the quinazoline **58**.

The chlorination of compound **58** with phosphorus oxychloride afforded compound **59,** which was reacted with the aniline derivative **60** to give compound **61**, as shown in Figure 8. Compound **61** underwent a regioselective arylation with furfural **62** catalyzed by palladium acetate in the presence of tricyclohexylphosphine tetrafluoroborate (Cy3P.HBF4) ligand to give **63**, which underwent a reductive amination with NaBH(OAc), followed by a reaction with *p*-toluenesulfonic acid to give lapatinib ditosylate monohydrate. 

##### Target Kinases 

Rusnak et al. [104] evaluated the inhibitory activity of lapatinib against a purified EGFR, ErbB-2, and ErbB-4. The results revealed potent inhibitory activities against the EGFR and ErbB2, with IC_50_ values of 9.8 and 10.2 nM, respectively. The inhibitory activities of lapatinib against the tested kinases expressed in IC_50_ values are presented in Figure 40. 

The effect of lapatinib on thymidylate synthase was investigated by Kim et al. [105]. The study revealed the inhibition of the nuclear translocation of the EGFR and HER2 and the downregulation of thymidylate synthase. 

##### Crystal Structures and Binding Interactions 

Lapatinib exists in two crystal structures in the protein data bank. These crystals include the complex of lapatinib with EGFR kinase domain (pdb: 1XKK) [106], and ErbB-4 kinase (pdb: 3BBT) [107]. The visualization of the binding mode/interactions of lapatinib into the kinase domain of the EGFR (pdb: 1XKK) is illustrated in Figure 41. 

##### Biological Activity 

Rusnak et al. [104] evaluated the efficacy of lapatinib against several cancer cell lines, including N5 (head and neck), A-431 (vulva), BT474 (breast), CaLu-3 (lung), and N87 (gastric). These cancer cell lines overexpress the EGFR or ErbB-2. The results revealed average IC_50_ values lower than 0.16 μM. 

Lapatinib was also evaluated against breast cancer BT474 cells overexpressing HER2 [108]. The results revealed a growth inhibitory activity at an IC_50_ value of 100 nmol/L. Lapatinib also showed inhibitory activity against the gastric cancer cells [109].

The ability of lapatinib to target the EGFR and HER2 leads to the prevention of autophosphorylation and the activation of oncogenic intracellular signaling pathways [110]. Accordingly, lapatinib was approved for metastatic breast cancer overexpressing HER-2 [23]. 

##### Metabolism

Castellino et al. [111] investigated the metabolism of [^14^C] lapatinib in human volunteers. The study revealed that the elimination of lapatinib takes place predominantly through feces (92%). The seven metabolites identified in the plasma were formed through *N*- and α-carbon oxidation, while the fecal metabolites were proposed to be formed through oxidation (*N*- and α-carbon), oxidative defluorination, and formation of the hydroxypyridine, as shown in Figure 42. 

Among the proposed metabolites, pyridinium salt M5 was formed from the bioactivation of the furane ring, an intramolecular cyclization. The hydroxypyridine metabolite M2 was formed from M5 after the loss of the ethyl sulfone moiety from M5.

In addition, M8, a hydroxylamine metabolite, was also identified in plasma. The oxidation of M8 afforded the regioisomers M6 and M7 [111]. In addition, the hydrolysis and oxidation of M7 produced the two geometric oxime isomers (M9 and M10). The *O*-dealkylation of the 3-fluorobenzyl moiety afforded the M1 with the phenolic hydroxy group. In addition, the *N*-dealkylation of 2-(methylsulfonyl)ethanamino moiety in lapatinib afforded the aldehyde M11, which underwent oxidation to the carboxylic acid M12. 

On the other hand, several synthetic routes of the radiolabeled lapatinib were also reported, which can be used in biodistribution and pharmacokinetic studies [112,113]. 

Nunes et al. [112] also reported a facile synthesis of ^18^F-labeled lapatinib that can be used in pharmacokinetic evaluation. In addition, Saleem et al. [113] reported the synthesis of [^11^C] lapatinib-PET. 

#### 1.2.9. Neratinib

##### Approval History

Neratinib (Figure 43) is an irreversible inhibitor of the EGFR, HER2/4 receptor tyrosine kinase, which was approved on 17 July 2017 for use in the treatment of HER2-positive breast cancer [114]. In 2020, it was also approved for HER2-positive metastatic breast cancer [115]. 

##### Synthesis 

The synthesis of neratinib was reported by Tsou’s group in a three-step method with overall yield of 47.3% [116]. However, several synthetic routes were reported with higher overall yield or with different reagent and/or reaction conditions [52]. Among these methods, Gu et al. [117] described a new synthetic route of neratinib using the Wittig–Horner reaction, as shown in Figure 9.

The substituted aniline **67** was obtained by coupling **65** with **66** and the reduction of the nitro group, as shown in Figure 9. The coupling between compound **68** and **67** afforded compound **69**, which was reacted with ethyl diethoxyphosphinyl acetate **70** to give **71**. The reaction of 71 with diethylacetal **72** afforded neratinib [117]. 

Line 748–749 indicate that compound 67 was obtained by coupling 65 with 66 and by reducing the nitro group to amino by Fe and acids (two steps)!

##### Target Kinases 

The kinase inhibitory activity of neratinib was evaluated by Tsou et al. [116]. The results revealed the inhibition of the EGFR and HER2 at IC_50_ values of 0.092 and 0.059 nM, respectively. 

Rabindran et al. [118] evaluated the kinase inhibitory activity of neratinib against ErbB family of receptor TK. The results revealed inhibitory activities at IC_50_ values of 92 and 59 nM against the EGFR and HER2, respectively, as shown in Figure 44. In addition, neratinib showed inhibitory activity against HER-4. On the other hand, very weak inhibition was observed against KDR, Akt, and CDKs.

Because of its multikinase inhibitory activity, neratinib was evaluated in several reports using different assay conditions. Accordingly, some differences in the reported IC_50_ values against the ErbB family were observed. The inhibitory activity of neratinib against the EGFR, HER2, and HER4 was observed at IC_50_ values of 1, 6, and 2.4 nM, respectively [110]. In another study [119], neratinib also showed IC_50_ values of 1.8 and 5.6 nM against the EGFR and HER-2, respectively. 

##### Crystal Structures and Binding Interactions 

Neratinib exists in two crystal structures in the protein data bank. These crystal structures include the complex of neratinib into the kinase domain of the EGFR T790M mutation (pdb: 2JIV) and the EGFR T790M/L858R mutant (pdb: 3W2Q) [120,121].

The binding interactions of neratinib into the active site of the EGFR T790M kinase domain (pdb: 2JIV) are visualized in Figure 45. One hydrogen bond between neratinib and Met793 could be observed. In addition, the binding interactions include one carbon hydrogen bond with Gln791, one halogen interaction with Ile744, and two electrostatic interactions with Asp855 and Met790. 

The binding mode and interactions of neratinib into the active site of the EGFR kinase domain T790M/L858R mutant are illustrated in Figure 46. Neratinib also showed one conventional hydrogen bond with Met793 and two electrostatic interactions with Glu762 and Met790. 

##### Biological Activity 

Rabindran et al. evaluated the effect of neratinib on the proliferation of several cancer cell lines [118]. The survival of the cell lines was evaluated using a protein binding dye assay after 24 h treatment of the tested cancer cell lines with neratinib, while BT 474 cells were treated for 6 days. The results revealed potent inhibitory activity against the cell lines which express high level of HER-2. Neratinib exhibited IC_50_ values of 2 and 3 nM against SK-Br-3 and 3T3/neu, respectively, as shown in Figure 44. Neratinib also exhibited inhibitory activity against 3T3, A431, MDA-MB-435, and SW620 with IC_50_ values in the range of 81–960 nM. 

##### Metabolism

The metabolism of neratinib was evaluated both in vivo and in vitro by Liu et al. [122]. The results revealed the identification of 12 metabolites. Among these metabolites, M6 and M7 were identified as major metabolites, which indicated that the GSH conjugation of neratinib is a major metabolic pathway, as shown in Figure 47. The metabolic pathways also included the *O*-dealkylation of the pyridin-2-ylmethyl moiety which gives M3. In addition, the formation of M12 and M10 was suggested to take place through *N*-oxidation and *N*-demethylation of neratinib, respectively. 

#### 1.2.10. Olmutinib

##### Approval History

Olmutinib (Figure 48) is a third-generation EGFR-TKI that was approved in May 2016, in South Korea for use in the treatment of patients with NSCLC with a T790M-positive mutation [123]. 

##### Synthesis

2,4-Dichlorothieno [3,2-d]pyrimidine **73** was used in the original synthesis of olmutinib, which is characterized by a very low yield [52]. Flick et al. [124] described the most likely synthetic approaches of olmutinib, which could be obtained from compound **73** in two steps, Figure 10. In the first step, compound **75** could be prepared from the reaction of 2,4-dichloro-thieno [3,2-*d*]pyrimidine **73** with the acrylamide derivative **74**. The reaction of **73** and **74** in DMF in the presence of potassium carbonate takes place with complete regioselectivity. The reaction of compound **75** with 4-(4-methylpiperazin-1-yl)aniline **76** afforded olmutinib. 

##### Target Kinases

Olmutinib exhibited inhibitory activities against selected lung cancer cell lines [125]. In vitro, olmutinib showed inhibitory activities against H1975 (L858-T790M) and HCC827 (exon 19 del.), and H358 (wild type EGFR NSCLC) at IC_50_ values of 9.2, 10, and 2225 nM, respectively, as shown in Figure 49. 

In another study, olmutinib also showed inhibitory activity against the EGFR (L858R/T790M) at an IC_50_ value of 10 nM [126]. On the other hand, olmutinib inhibited the activity of PI3Kα at an IC_50_ value > 10 μM.

##### Crystal Structures and Binding Interactions 

Olmutinib was not reported in any crystal structure in the protein data bank. In addition, limited studies investigated its binding mode and interactions. Of these studies, the binding mode of olmutinib into EGFR T790M was investigated in a docking study by Hu et al. [126]. 

##### Biological Activity 

The cytotoxic activity of olmutinib was also evaluated against a panel of four cancer cell lines and one normal cell line [126]. The results revealed IC_50_ values of 4.29 μM (A549), 0.52 μM (H1975), 5.29 μM (NCI-H460), 26.90 μM (MCF-7), and 25.76 μM (LO2). 

In addition, olmutinib was also evaluated for multidrug reversal activity in cancer cell lines overexpressing ABCB1, ABCG2, or ABCC1 transporters [127]. The results revealed the ability of olmutinib to reverse the resistance mediated by ABCG2 only. 

##### Metabolism 

The pharmacokinetic analysis of olmutinib in healthy volunteers was investigated by Noh et al. [128]. The results of this study indicated a potential role of glutathione S-transferase in the metabolism of olmutinib. 

Attwa et al. also investigated the phase I metabolism of olmutinib in rat liver microsomes [129]. The study was performed to investigate the formation of reactive metabolites. The results revealed that the hydroxylation of the piperazine ring in olmutinib is the major metabolic pathway. In addition, seven reactive metabolites were identified in this study, as shown in Figure 50. 

#### 1.2.11. Osimertinib

##### Approval History

Osimertinib (Figure 51) is a third-generation, irreversible EGFR-TKI that was approved by the FDA on November 13, 2015 for use in the treatment of EGFR T790M mutation-positive NSCLC [130]. On 21 December 2020, it was approved as an adjuvant treatment of NSCLC patients with early-stage EGFR mutation. 

##### Synthesis

Finlay et al. reported the original synthesis of osimertinib [131]. However, this synthetic route consists of seven steps with a low overall yield, which encouraged the researchers to discover better synthetic routes. Zhu et al. [132] reported a new and convergent synthetic route in which osimertinib was prepared in six steps with an overall yield of 40.4%, as shown in Figure 11. 

The synthesis of the key intermediates **80** and **83** was achieved according to Figure 1. The reaction of **80** and **83** in 1-butanol afforded compound **84**. The reduction of the nitro group in **84** was achieved using H_2_/Raney Nickel. The acylation of the amino group in compound **85** with acryloyl chloride gave osimertinib.

##### Target Kinases

The inhibitory activity osimertinib was investigated against the wild-type and mutant variants of the EGFR [40]. The results revealed IC_50_ values in the range of 3–13 nM against the four EGFR mutations, compared to 938 nM against the wild-type EGFR, as shown in Figure 52. 

The high potency of osimertinib could be attributed to the covalent bonding with EGFR, which leads to the irreversible inhibition of kinase activity [133]. In another study, osimertinib showed inhibitory activities against the EGFR (L858R), EGFR (d746–750), EGFR (L861Q), EGFR (L858R/T790M), and EGFR (d746–750/T790M) at IC_50_ values in the range of 0.1–4 nM, compared to 30 nM against the wild-type EGFR [134]. 

##### Crystal Structures and Binding Interactions

Osimertinib exists in three crystal structures in the protein data bank. The crystal structures include the complex of osimertinib with the wild-type EGFR (pdb: 4ZAU) [135], EGFR 696-1022 T790M (pdb: 6JX0) [136], and EGFR L858R/T790M/C797S (pdb: 6LUD) [137].

The binding mode of osimertinib in the wild-type EGFR is illustrated in Figure 53. Osimertinib showed three conventional hydrogen bonds with Met793 and Cys797 [135].

The binding mode and interactions of osimertinib into the active site of the EGFR 696-1022 T790M (pdb: 6JX0) are visualized in Figure 54. Osimertinib shows two conventional hydrogen bonds into the active site of EGFR 696-1022 T790M [136]. 

The binding mode and interactions of osimertinib into the active site of the EGFR L858R/T790M/C797S are illustrated in Figure 55. Osimertinib displayed two hydrogen bonds with Met793 and Ser7979 amino acids. 

##### Biological Activity

Gao et al. [138] investigated the antiproliferative activity of osimertinib against three cancer cell lines using an MTS assay. The results revealed antiproliferative activity against NCI-H460 (IC_50_ = 415.9 nM), PC9 (IC_50_ = 6.5 nM), and NCI-H1975 (IC_50_ = 10.5 nM). 

The antiproliferative activity of osimertinib was also evaluated against a panel of lung cancer cell lines, including PC-9 (exon 19del), H3255 (L858R), PC-9ER (exon 19del + T790M), H1975 (L858R + T790M), and BID007 (A763-Y764insFQEA). The results revealed the inhibition of proliferation in the cancer cell lines at IC50 values in the range of 4–40 nM [40].

##### Metabolism

Finlay et al. [131] reported the formation of two *N*-dealkylated metabolites (AZ5104 and AZ7550) formed by the dealkylation of the indole *N*-methyl or the demethylation of the terminal amine in the *N,N,N*′-trimethylethylenediamine side chain. The two metabolites showed inhibitory activity against the wild-type and mutated EGFR with kinase selectivity compared to or lower than the parent, osimertinib [131,133]. 

Dickinson et al. [139] also investigated the metabolism of osimertinib using [^14^C] osimertinib. The results revealed the formation of seven metabolites in human hepatocytes, as shown in Figure 56. Among the metabolites detected in human plasma, AZ7550 and AZ5104 were the most abundant, while AZ5104 was the most abundant metabolite in feces. 

#### 1.2.12. Pyrotinib

##### Approval History

Pyrotinib (Figure 57) is an irreversible ErbB-TKI that acts through the inhibition of EGFR, HER2, and HER4 receptor tyrosine kinase activity. It was approved in China in 2018 for use in the treatment of HER2-positive breast cancer [140]. 

##### Synthesis

The synthesis of the pyrotinib (Figure 12) was started by the reduction of the pyrrolidine-1-carboxylate **86** with lithium aluminium hydride to give compound **87** [141]. The (2*R*)-1-methylpyrrolidine-2-carbaldehyde **88** was prepared from the reaction of **87** and oxalyl chloride. The acylation of the amino group in **89** with diethyl phosphonoacetic acid using CDI as a coupling catalyst afforded **90**, which on the reaction with **88** afforded pyrotinib as a free base. 

##### Target Kinases

Pyrotinib and three of its principal metabolites (M1, M2, and M5) were evaluated in regard to their kinase inhibitory activities [141]. The results revealed that pyrotinib inhibits the EGFR and HER2 kinases at IC_50_ values of 13 and 38 nM, respectively. In addition, the three metabolites also exhibited inhibitory activity against EGFR kinases at the nanomolar level (IC_50_ = 2.4–283.1 nM), while weaker activities were observed against HER2. Among the three metabolites, M1 inhibited HER2 at an IC_50_ value of 220.2 nM, as shown in Figure 58. 

##### Crystal Structures and Binding Interactions

No crystal structures of pyrotinib with any of its target kinases were reported in the protein data bank. However, the proposed binding mode of pyrotinib into HER2kinase was investigated in a docking study, which revealed one hydrogen bond with Met801 [141].

##### Biological Activity

The anticancer activity of pyrotinib against two breast cancer (BT474 and MDA-MB-231) cell lines and one ovarian SK-OV-3 cancer cell line [141] was examined. The results revealed antiproliferative activity against BT474, SK-OV-3, and MDA-MB-231 cancer cell lines at IC_50_ values of 5.1, 43.0, and 3500 nM, respectively. These results indicated the high antiproliferative activity of pyrotinib against HER2-dependant cancer cell lines. 

In addition, the combination of pyrotinib and apatinib showed a synergetic anticancer effect in HER2-positive NCI-N87 xenografts [142]. The study also revealed the re-sensitization of pyrotinib-resistant cells to pyrotinib when combined with apatinib. 

##### Metabolism 

The human metabolism of pyrotinib was investigated by Zhu et al. [143]. The results revealed the identification of 24 metabolites. These metabolites were classified into two groups. The first includes 16 of phase I metabolites, while the second group includes 8 of phase II. The study also revealed the formation of three principal metabolites, M1, M2, and M5. The three metabolites were produced from the *O*-dealkylation of picoline moiety and oxidation of the pyrrolidine ring into a lactam ring, as shown in Figure 59.

Meng et al. [144] also investigated the metabolism of pyrotinib in healthy male Chinese volunteers to identify the potential covalent binding interactions with plasma protein. The study was performed using [^14^C] pyrotinib which was incubated with plasma, serum albumin, or α1-acid glycoprotein. The results revealed that 22.0–53.3% of pyrotinib binds covalently to the protein. The study also identified the adducts formed between [^14^C] pyrotinib and the plasma protein. These adducts included the adducts formed with lysine, glycine-lysine, Gly-Lys-Ala, and Lys-Ala-Ser. A general structure of the pyrotinib-protein adducts is presented in Figure 60.

#### 1.2.13. Simotinib

##### Approval History

Simotinib (Figure 61) is a selective EGFR-TKI. It was approved in China in 2018 for use in the treatment of solid tumors. It is indicated for metastatic NSCLC [145].

##### Synthesis 

The synthesis of simotinib was described in the patent application (US 2007/0167470 A1) [146]. Due to the high structural similarity between simotinib and gefitinib, the two compounds shared the same intermediate **45** in their synthetic routes, as shown in Figure 6. The alkylation of compound **45** by 3-bromopropanol **91** gave **92**, which could afford simotinib on the reaction with 5,8-dioxa-10-azadispiro[0,2,3,4]undecane, as shown in Figure 13.

##### Target Kinases

Mechanistic studies of simotinib showed the selective and dose-dependent inhibition of the EGFR with an IC_50_ value of 19.9 nM [147].

##### Crystal Structures and Binding Interactions

Simotinib was not reported until now in any crystal structure in the protein data bank.

##### Biological Activity

The cytotoxic activity of simotinib was assessed in vitro against a panel of cancer cell lines [148]. The results revealed inhibitory activity against A549, MCF-7, and SGC7901 cells at IC_50_ values of 2.2 μM, 12.6 μM, and 10.6 μM, respectively.

The results of the pharmacokinetic study of simotinib in healthy volunteers showed an average clearance (CL) *T*_1/2_ of 8–12 hrs [147]. Phase Ib of simotinib was also performed in 41 patients with *EGFR* gene mutations [147]. The results of this study revealed that simotinib was well tolerated at a dose up to 650 mg.

Li et al. [149] also developed a UPLC–MS/MS assay method to study the pharmacokinetics of simotinib. In addition, Han et al. [150] also studied the pharmacokinetics of simotinib in NSCLC patients. The results revealed that after a single dose, simotinib reaches a T_max_ of 1–6 h and a half-life of 4.7–11.1 h.

Zhu et al. investigated the drug–drug interactions of simotinib [151]. The study revealed the ability of simotinib to increase the paracellular permeability of intestinal epithelial cells, which leads to an upregulation of the intestinal absorption.

The effect of simotinib on P-gp-associated MDR was investigated by Huang et al. [148]. The results revealed that simotinib could reverse the P-gp-mediated MDR, although it is a weak substrate of P-gp. Mechanistic studies of simotinib revealed the inhibition of rhodamine 123 efflux and an increase in the intracellular concentration of the anticancer agents. Accordingly, it decreased the IC_50_ value of paclitaxel and adriamycin in A549 cells by 2.23- and 3.92-fold, respectively [148].

##### Metabolism

The safety and pharmacokinetics of simotinib was evaluated in a Phase I clinical study in patients with NSCLC (ClinicalTrials.gov identifier: NCT01772732) [147]. The results revealed that at a dose of up to 650 mg, simotinib was well tolerated. In another study, simotinib showed a half-life (T_1/2_) consistent with icotinib [150].

#### 1.2.14. Vandetanib

##### Approval History

Vandetanib (Figure 62) is a multikinase inhibitor of angiogenesis and cancer cell proliferation [152]. It was first approved by the FDA on 6 April 2011 for use in treating patients with metastatic medullary thyroid cancer [153].

##### Synthesis

Several methods were reported that described the synthesis of vandetanib using different reagents and reaction conditions [93,154,155]. Among these was the method reported by Brocklesby et al. [155] which mainly depends on the microwave-accelerated Dimroth rearrangement, as shown in Figure 14.

The synthesis of vandetanib started with compound **94**, where the phenolic OH group was protected using benzyl chloride in the first step, as shown in Figure 14. The nitration of compound **95** followed by the reduction of the nitro group in **96** using sodium dithionate afforded **97**.

The reaction of compound **97** with dimethylformamide-dimethylacetal (DMF-DMA) for 15 min of microwave irradiation afforded **98** in low yield [155]. However, this yield was improved by increasing the time of the microwave irradiation. Compound **98** underwent debenzylation using TFA to give compound **99**. The OH group in compound **99** was alkylated by *tert*-butyl-4-(tosyloxy)methyl)piperidine-1-carboxylate **100** to give compound **101**. The reaction of compound **101** with the substituted aniline **102** takes place via Dimroth rearrangement to give compound **103**. The deprotection of Boc moiety in compound **103** gave **104**, which underwent reductive amination with formaldehyde to give vandetanib, as shown in Figure 14.

##### Target Kinases

Investigation of the mechanism of action of vandetanib revealed multi-kinase inhibitory activity [156,157]. Vandetanib inhibited the activity of VEGFR-1/2/3 at IC_50_ values in the range of 0.04–1.60 μM. In addition, it inhibited the EGFR at an IC_50_ value of 0.5 μM. Vandetanib also inhibited the activity of the oncogenic RET kinase at an IC_50_ value of 0.13 μM, as shown in Figure 63.

##### Crystal Structures and Binding Interactions

Vandetanib is available as a co-crystallized ligand in the tyrosine kinase domain of RET kinase (pdb: 2IVU) [158]. The binding modes of vandetanib (Figure 64) show one conventional hydrogen bond with Ala807 and three carbon hydrogen bonds with Glu805, Tyr806, and Gly810. Besides the hydrophobic interactions, vandetanib also exhibited one electrostatic interaction of the pi-cation type with Lys758 and one halogen interaction with Ala756, as shown in Figure 64.

##### Biological Activity

The cytotoxicity of vandetanib was investigated against a wide range of cancer cell lines [159]. The results revealed growth inhibitory activity against PC-9 and PC-9/ZD lung cancer cell lines at IC_50_ values of 0.14 and 5.92 μM, respectively. In addition, vandetanib showed antiproliferative activities against TKKK cells, TGBC24TKB, OZ, and HuCCT1 cell lines at IC_50_ values of 0.22, 4.5, 12.2, and 10 μM, respectively [160].

The anti-breast cancer activity of vandetanib was investigated by Sarkar et al. in MCF-7, T-47D, ZR-75-1, and MDA-MB-231 cancer cell lines [161]. The results revealed the ability of vandetanib to inhibit the growth and induced cell cycle arrest in the cancer cell lines. In addition, vandetanib inhibited the growth of human MDAMB-231 cells in a xenograft model.

##### Metabolism

The pharmacokinetics of vandetanib in healthy subjects were studied by Martin et al. [162] using [^14^C] vandetanib. The study revealed that 25% of the given dose was recovered in urine, while 44% was recovered in feces. In most of the urine samples, vandetanib–glucuronic acid (GU) conjugate was found. In addition, analysis of the plasma, urine, and feces revealed the presence of the vandetanib, *N*-oxide and *N*-desmethyl metabolites, as shown in Figure 65.

Indra et al. [163] also studied the metabolism of vandetanib to identify the human oxidizing enzymes involved in the metabolism. The results revealed a correlation between the formation of the *N*-desmethylvandetanib/vandetanib-*N*-oxide and the activities of CYP3A4/flavin-containing monooxygenases (FMO). In addition, the results revealed that cytochrome b5 stimulates the CYP3A4-catalyzed formation of *N*-desmethylvandetanib. Indra and his coworkers also performed a molecular docking study which revealed binding of more than one molecule to the active center of CYP3A4, justifying the high efficiency of this enzyme in converting vandetanib to the *N*-demethylated product.

The formation of the reactive intermediates on the metabolism of vandetanib was investigated by Attwa et al. [164]. The drug was incubated with rat liver microsomes, where the results revealed the identification of several metabolic pathways, as shown in Figure 66. The study also revealed that the *N*-methyl piperidine moiety can form several reactive iminium intermediates which were trapped by potassium cyanide. These intermediates could account for the idiosyncratic toxicities of vandetanib. In addition, vandetanib–glucuronic acid conjugate, a phase II metabolite, was also identified in this study.

## 2. Conclusions

Targeting the tyrosine kinase domain in wild-type and mutants of the EGFR with small-molecule inhibitors is confirmed as a valid strategy in cancer chemotherapy. Since the approval of the first EGFR-TKI, erlotinib, considerable effort has been devoted to the discovery and development of new potent and safe inhibitors. Fourteen EGFR small-molecule inhibitors have been approved so far for the treatment of different types of cancers. The primary focus of this review was on EGFR-TKIs, which have been approved for the treatment of different types of cancers. These drugs were classified based on their chemical structures, target kinases, and pharmacological uses. In addition, the synthetic routes to each of these drugs were discussed. The binding modes and interactions of these drugs into their target kinases were visualized and discussed. Based on the nature of the binding interactions with the target kinases, these drugs may be classified as reversible or irreversible inhibitors. The cytotoxic activities of these drugs against different types of cancer cell lines were also summarized. In addition, the metabolic pathways and the various metabolites associated with the fourteen drugs were also presented. However, the effectiveness of these drugs is challenged by the development of single, double, and triple mutations in the EGFR. Recent results from preclinical and clinical studies of fourth-generation EGFR-TKIs indicate these may provide effective treatments for patients with such mutations. Overall, this review highlighted the syntheses, target kinases, crystal structures, binding interactions, cytotoxicity, and metabolism of the fourteen approved EGFR inhibitors. These data should greatly help in the design of new EGFR inhibitors.

## 3. Perspective

Currently, first- to third-generation EGFR-TKIs are in clinical use for the treatment of different types of cancers. Although high treatment efficacy is established, the development of several mutations within the EGFR [165] affects drug response rates in NSCLC patients [166]. The EGFR C797S mutation has rapidly led to resistance to third-generation EGFR-TKIs [33], such that these are becoming ineffective in patients with these mutations [32]. Accordingly, the primary focus of the research in this area is directed toward the discovery and development of new therapeutics that can target EGFRs carrying the del19/L858R+ C797S ± T790M mutations. Recently, fourth-generation allosteric mutant-selective EGFR inhibitors have been developed that show synergistic anticancer effects against NSCLC with a mutant EGFR when combined with traditional EGFR inhibitors [32]. In addition, several fourth-generation EGFR-TKIs (EAI001, EAI045, BLU-945, and BBT-176) have been investigated against cancer cells with double or triple EGFR mutation [32,33,34,35]. One of these compounds, BBT-176, is in a Phase I/II trial in NSCLC patients with advanced lung cancer [35]. Should one of these drugs at least reach the market, this will greatly encourage further research in this area.

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
