# Peer review of "Globally Approved EGFR Inhibitors: Insights into Their Syntheses, Target Kinases, Biological Activities, Receptor Interactions, and Metabolism"

_molecules, 2021, doi:10.3390/molecules26216677_

Round 1

Reviewer 1 Report

             The review "Globally approved EGFR inhibitors: insights into their syntheses, target kinases, biological activities, receptor interactions, and metabolism" by Mohammed Abourehab, Alaa Alqahtani, Bahaa  M. Youssif and Ahmed  Gouda presents an overview on the synthesis, mode of action, and binding of clinically approved small-molecule EGFR inhibitors. The review commences with a brief commentary on the structure and subunits of EGFR, its critical importance in acting as a receptor for EGF, and subsequent mediation of signalling cascades that control cell proliferation and apoptotic events. Structures of the approved EGFR are given, followed by an exacting classification into subtypes based on chemical structures, the nature of their interaction, whether reversible or irreversible, with the various active sites associated with the mode of inhibition within EGFR, whether these inhibition modes are specific, or as may be associated with other targets, or as associated with their clinical use, and finally generational aspects as may be driven by structural changes due to mutations in the original binding sites. Thereafter, EFGR inhibitors approved for treatment of cancer are discussed in terms of their registration history, synthesis, activity on target kinases, and details of published structures and binding interaction, and then their biological activities and their metabolism.

             Overall, this has to be rated as a notably comprehensive review, and in this sense, it does stand out from rather a lot of other very recent reviews on this intensively reviewed topic. The authors have to be congratulated on their efforts, and the general impression is that once fully polished, the review will attract a wide readership; with its chemical aspect, this should have a central appeal to medicinal chemists. However, at the moment, the MS is uneven in quality in relation to the appearance of the figures, schemes and flow of the text - a lot will need to be done to render the manuscript suitable for publication. The MS must be corrected before the work can be considered suitable for publication. Whilst it is not possible to list all aspects, the following points may be used as examples.

Comments: Factually, structures of the subject molecules appear to be correct, although there are mistakes in the figures and schemes.  The use of the descriptor 'kinase' should be carefully exercised within the scope of its definition. Where acronyms first appear, e.g. EGF, these should be defined, and not later in the text.  There are a number of spelling and typographical errors, errors of grammar, e.g. use of wrong tense in sentences, and wrong descriptors and use of incorrect definitions that detract from the quality of the MS. Some such mistakes raise questions over the standard of proof reading before the MS was submitted.  There are too many mistakes to highlight, but examples are included in the following, which hopefully should be sufficient to show what is required:

  1. Abstract: errors of grammar detract, including use of wrong tense in verbs to indicate a temporal aspect, e.g. as an example, from line 20, better written as something like: "In this review, we focus on the EGFR small-molecule inhibitors that have been approved for clinical use in cancer therapy. The drugs are classified according to their chemical structures, target kinases, and pharmacological uses.  The syntheses of these drugs are also discussed.  The crystal structures of the drugs with their target kinases are also discussed and their binding modes and nature of their interactions are visualized. The data in this work should assist in the design of potent new EGFR inhibitors."
  2. Lines 40-43: better as "EGFR is composed of three subunits (Fig. 1). The first is the extracellular EGF-binding domain, the second is the transmembrane domain, and the third the cytoplasmic domain [2]. The cytoplasmic domain consists of two subunits, the protein tyrosine kinase (PTK) domain, and the C-terminal phosphorylation domain."
  3. Figure 1 closely resembles that in other reviews; in any event, even though the some of the acronyms are defined in the ensuing text, to make it more useful to the reader, each of the signalling pathways should be defined in the figure caption, e.g. PI3K, AKT, mTOR etc.
  4. Lines 58-64: N-lobe and C-lobe should be the N-loop and C-loop, respectively; please define AMPPNP; is visualized. Presumably the authors used Discover Studio Visualizer to generate the binding modes in Figure 2; this should be stated in the caption to this, and in the captions to ALL figures where this program is used.
  5. Figure 2 and ALL figures where Discover Studio Visualizer is used: descriptions of the interaction in the B figure are too difficult to read.
  6. Lines 73-75: Better as "These crystal structures provide a huge amount of information that enables the researcher in this field to understand the structure and function of EGFR. This must enhance the discovery and development of rationally designed EGFR tyrosine kinase inhibitors [17–19]".
  7. Figure 3: Change caption to "Chemical structures of the approved EGFR inhibitors". Whilst the structures have been checked by the reviewer, it is critical that the authors ensure that all structures are correct. In addition, whilst it is nice the heterocyclic core structures are depicted in a different colour to the appended substituents, the authors should ensure this is consistent throughout, e.g. structure of pyrotinib.
  8. Lines 93-94: Change to "….. the primary focus is on small-molecule inhibitors of EGFR which are approved for clinical use as anticancer agents."
  9. Line 99: Change to "Based on their chemical structures, the approved EGFR-TKIs can be …."
  10. Line 101: Change to "…., olmutinib is considered….."
  11. Figure 4: For the quinazoline-4,6-diamine subtype, the 6-amino group should be in blue color.
  12. Line 110: Change to "can also be classified as reversible or irreversible inhibitors (Fig. 5)."
  13. Line 110-112: Change to something like "For the first type, the inhibitors competitively bind to the ATP binding site in EGFR through noncovalent interactions involving electrostatic, hydrogen-bonding, and hydrophobic interactions"
  14. Lines 114-116: Is it true that these irreversible inhibitors always react with the cysteine residues in EGFR? In any event, change the last sentence to something like "The structure of these inhibitors is characterized by the presence of an electrophilic side chain that acts as a Michael acceptor which chemically reacts with the cysteine thiol group to form the addition product."
  15. Figure 5: Change caption to "Classification of EGFR-TKIs based on nature of inhibition of EGFR".
  16. Lines 121-123: Change to "The EGFR inhibitors can also be classified into three groups based on the treatment of the types of cancers for which they are approved [26]. The first group includes the drugs which are approved for the treatment of NSCLC such as afatinib….." <Note that the acronym NSCLC must be defined where it first appears - non-small-cell lung carcinoma>.
  17. Line 144: What is the advantage of the second-generation EGFR-TKIs over the first generation EGFR-TKIs?
  18. Lines 150-151: Change to something like "These moieties bear an acrylamide group that as noted above forms a covalent bond with the cysteine residue in EGFR" <use of moiety in this specific context is not correct; group is much better>
  19. Lines 156-159: Change to something like "The rapid emergence of EGFR T790M and C797S mutations has led to the development of resistance to the third-generation EGFR-TKIs [30]. <the authors may like to note the C797S mutation deprives the ability of the 3rd gen TKIs to act as irreversible inhibitors, as the Michael donor group – the SH in cysteine is replaced by the ineffective –OH in serine>. In addition, the emergence of EGFR double and triple (del19/L858R + C797S ± T790M) <this should be expanded upon and explained more carefully> mutations challenges the therapeutic effectiveness of EGFR-TKIs. These problems underscore the ongoing need to develop new and potent EGFR-TKIs."

Lines 161-164: the development of the allosteric EGFR inhibitors is a noteworthy achievement, although it is recognized that it would not be easy to incorporate structural aspects into the current review; change to "Recently, several fourth generation allosteric EGFR inhibitors that bind to a site in EGFR other than the PTK domain were reported [32]. However, although these inhibitors are ineffective against NSCLC with mutated EGFR, they display synergistic anticancer effects when combined with the current EGFR inhibitors" <<will help to indicate which ones>>

  1. Lines 165-166: best to combine this paragraph with the preceding paragraph and as structures for EAI045, BLU-945, and BBT-176 are not given, perhaps best to change to something like "Currently, several fourth generation EGFR-TKIs are being evaluated for efficacy against cancers carrying double or triple EGFR mutations [33–35]. One of these compounds, BBT-176 is in Phase I/II trials in NSCLC patients with advanced lung cancer [35], but these inhibitors are not yet approved for clinical use." Otherwise insert a figure showing structures for these fourth generations TKIs, and leave descriptors in the text.
  2. The next part of the review comprises descriptions of the individual EGFR inhibitors, with overviews of the approval history, synthesis, target kinases, details of the inhibitor binding, biological activity, and metabolism. The presentation of such data overall represents the main strength of this review, and is richly informative. However, the authors should amend and improve these sections according to the foregoing.
  3. The Schemes overall need to be visually enhanced; at the moment the diagrams and reaction arrows are not so attractive, the captions should be consistent, e.g. Scheme 1. Synthesis of afatinib, and in certain cases there are mistakes.
  4. Scheme 1: hashed bond in compound 4 and subsequent needs to be displayed properly.
  5. Lines 195-196: Change to "Afatinib inhibits ……..In addition, it shows …" <here and in numerous places in the following text when describing efficacies as reflected in IC50 and other data>
  6. Line 211 and Figures 9, 10: Did the authors use Discover Studio Visualizer to generate the binding modes in Figures 9 and 10?; if so, this should be stated in the caption to this, and ALL subsequent figures where this program is used.
  7. Lines 233-234: Change to "The formation of this covalent bond leads to irreversible inhibition of the autophosphorylation of the ErbB family [41]. The mechanism of the irreversible inhibition of EGFR by afatinib is illustrated in Fig. 11."
  8. Lines 238-242: Change to something like "Afatinib binds to and inhibits EGFR with exon 19 deletion mutations and exon 21 L858R mutations. The covalent interaction leads to irreversible inhibition of the kinase activity which provides an advantage over other non-covalent competitive inhibitors of EGFR such as erlotinib and gefitinib [43–45]."
  9. Line 258 Metabolism - here and for all subsequent compounds these sections have to be improved, particularly with regard to the Figures illustrating the metabolic conversions. Thus in Figure 12 the caption must be changed from "The main metabolic of afatinib must be changed to something like "the principal metabolic pathway for afatinib". The Figure 12 is poorly presented: the reaction indicating addition of 'RH' has to be changed to 'RSH' and the curved arrows changed to indicate the correct chemical sense (cf. Figure 11 displaying addition of Cys-797); also one presumes that the authors know the reaction is reversible. The statement in lines 282-284 tends to indicate this is true.  Accuracy is important for ensuring credibility of a review.

28: Lines 277-284: There are spelling mistakes that must be corrected.

29: For the succeeding sections, the following are briefly noted.

  1. line 316, "The crystal structure of almonertinib bound to any of its target kinases has not yet been reported"; ii. structures in Figure 15 are incomprehensible even though a brief explanation is given below (lines 334-339); compare for example Figure 20, where structures are clearly shown iii. structures in Scheme 3 are not color-coded as in the preceding schemes, and others.
  2. The authors should use the foregoing to improve the manuscript overall.

Finally to help the authors, the Conclusions and Perspective may be written as follows – one does trust the meaning has not been changed from what the authors originally intended:

  1. Conclusion

             Targeting the tyrosine kinase domain in wild-type and mutants of EGFR with small-molecule inhibitors is confirmed as a valid strategy in cancer chemotherapy.  Since the approval of the first EGFR-TKI, erlotinib, considerable effort has been devoted to the discovery and development of new potent and safe inhibitors. Fourteen EGFR small-molecule inhibitors have so far been approved for the treatment of different types of cancers. The primary focus of this review is on EGFR-TKIs which have been approved for the treatment of different types of cancers. These drugs are classified based on their chemical structures, target kinases, and pharmacological uses. In addition, the synthetic routes to each of these drugs are discussed.  The binding modes and interactions of these drugs into their target kinases have been visualized and discussed. Based on the nature of the binding interactions with the target kinases, these drugs may be classified as reversible or irreversible inhibitors. The cytotoxic activities of these drugs against different types of cancer cell lines have also been summarized. In addition, the metabolic pathways and the various metabolites associated with the fourteen drugs are also presented. However, the effectiveness of these drugs is challenged by the development of single, double, and triple mutations in EGFR. Recent results from preclinical and clinical studies of fourth generation EGFR-TKIs indicate these may provide effective treatment for patients with such mutations. Overall, this review highlights the syntheses, target kinases, crystal structures, binding interactions, cytotoxicity, and metabolism of the fourteen approved EGFR inhibitors. These data should greatly help in the design of new EGFR inhibitors. 3. Perspective

             Currently, first- to third-generation EGFR-TKIs are in clinical use for the treatment of different types of cancers. Although high treatment efficacy is established, the development of several muations within EGFR [165] affect drug response rates in NSCLC patients [166]. The EGFR C797S mutation has rapidly led to resistance to the third generation EGFR-TKIs [33], such that these are becoming ineffective in patients with these mutations [32]. Accordingly, the primary focus of the research in this area is directed toward the discovery and development of new therapeutics that can target EGFR carrying the del19/L858R+ C797S ± T790M mutations. Recently, fourth generation allosteric mutant-selective EGFR inhibitors have been developed that show synergistic anticancer effects against NSCLC with mutant EGFR when combined with traditional EGFR inhibitors [32]. In addition, several fourth generation EGFR-TKIs (EAI001, EAI045, BLU-945 and BBT-176) are being investigated against cancer cells with double or triple EGFR mutation [32–35]. One of these compounds, BBT-176 is in Phase I/II trial in NSCLC patient with advanced lung cancer [35]. Should one of these drugs at least reach the market, this will greatly encourage further research in this area.

Author Response

Reviewer 1

Comments and Suggestions for Authors

             The review "Globally approved EGFR inhibitors: insights into their syntheses, target kinases, biological activities, receptor interactions, and metabolism" by Mohammed Abourehab, Alaa Alqahtani, Bahaa  M. Youssif and Ahmed  Gouda presents an overview on the synthesis, mode of action, and binding of clinically approved small-molecule EGFR inhibitors. The review commences with a brief commentary on the structure and subunits of EGFR, its critical importance in acting as a receptor for EGF, and subsequent mediation of signalling cascades that control cell proliferation and apoptotic events. Structures of the approved EGFR are given, followed by an exacting classification into subtypes based on chemical structures, the nature of their interaction, whether reversible or irreversible, with the various active sites associated with the mode of inhibition within EGFR, whether these inhibition modes are specific, or as may be associated with other targets, or as associated with their clinical use, and finally generational aspects as may be driven by structural changes due to mutations in the original binding sites. Thereafter, EFGR inhibitors approved for treatment of cancer are discussed in terms of their registration history, synthesis, activity on target kinases, and details of published structures and binding interaction, and then their biological activities and their metabolism.

             Overall, this has to be rated as a notably comprehensive review, and in this sense, it does stand out from rather a lot of other very recent reviews on this intensively reviewed topic. The authors have to be congratulated on their efforts, and the general impression is that once fully polished, the review will attract a wide readership; with its chemical aspect, this should have a central appeal to medicinal chemists. However, at the moment, the MS is uneven in quality in relation to the appearance of the figures, schemes and flow of the text - a lot will need to be done to render the manuscript suitable for publication. The MS must be corrected before the work can be considered suitable for publication. Whilst it is not possible to list all aspects, the following points may be used as examples.

Comments: Factually, structures of the subject molecules appear to be correct, although there are mistakes in the figures and schemes.  The use of the descriptor 'kinase' should be carefully exercised within the scope of its definition. Where acronyms first appear, e.g. EGF, these should be defined, and not later in the text.  There are a number of spelling and typographical errors, errors of grammar, e.g. use of wrong tense in sentences, and wrong descriptors and use of incorrect definitions that detract from the quality of the MS. Some such mistakes raise questions over the standard of proof reading before the MS was submitted.  There are too many mistakes to highlight, but examples are included in the following, which hopefully should be sufficient to show what is required:

Response: We highly appreciate the valuable comments of reviewer 1 and his valuable corrections that have been emerged after his careful and precise revision which would help in improving the quality of the manuscript. In addition, we indicated the revisions/corrections by a yellow highlighter in the revised manuscript. Below, are our responses to the comments, point-by point.

Comment: The while manuscript was revised and corrected for the typo and grammar mistakes

  1. Abstract: errors of grammar detract, including use of wrong tense in verbs to indicate a temporal aspect, e.g. as an example, from line 20, better written as something like: "In this review, we focus on the EGFR small-molecule inhibitors that have been approved for clinical use in cancer therapy. The drugs are classified according to their chemical structures, target kinases, and pharmacological uses.  The syntheses of these drugs are also discussed.  The crystal structures of the drugs with their target kinases are also discussed and their binding modes and nature of their interactions are visualized. The data in this work should assist in the design of potent new EGFR inhibitors."

Response: The abstract was revised and corrected per the reviewer’s comment

Comment:

  1. Lines 40-43: better as "EGFR is composed of three subunits (Fig. 1). The first is the extracellular EGF-binding domain, the second is the transmembrane domain, and the third the cytoplasmic domain [2]. The cytoplasmic domain consists of two subunits, the protein tyrosine kinase (PTK) domain, and the C-terminal phosphorylation domain."

Response: The syntax was modified per the reviewer’s suggestion

Comment:

  1. Figure 1 closely resembles that in other reviews; in any event, even though the some of the acronyms are defined in the ensuing text, to make it more useful to the reader, each of the signalling pathways should be defined in the figure caption, e.g. PI3K, AKT, mTOR etc.

Response: Figure 1 was updated. The acronyms were added to the caption of figure 1

Comment:

  1. Lines 58-64: N-lobe and C-lobe should be the N-loop and C-loop, respectively; please define AMPPNP; is visualized. Presumably the authors used Discover Studio Visualizer to generate the binding modes in Figure 2; this should be stated in the caption to this, and in the captions to ALL figures where this program is used.

Response:

The N- and C-lobes were revised in the cited reference (ref 12) and were kept as “N-lobe and C-lobe”

AMPPNP was defined in the text.

Discover Studio Visualizer was cited in all the figures, where it was used.

Comment:

  1. Figure 2 and ALL figures where Discover Studio Visualizer is used: descriptions of the interaction in the B figure are too difficult to read.

Response: The resolution of Fig. 2 and other figure was improved in the revised manuscript.

Comment:

  1. Lines 73-75: Better as "These crystal structures provide a huge amount of information that enables the researcher in this field to understand the structure and function of EGFR. This must enhance the discovery and development of rationally designed EGFR tyrosine kinase inhibitors [17–19]".

Response: The part was changed in the updated manuscript per the reviewer’s comment

Comment:

  1. Figure 3: Change caption to "Chemical structures of the approved EGFR inhibitors". Whilst the structures have been checked by the reviewer, it is critical that the authors ensure that all structures are correct. In addition, whilst it is nice the heterocyclic core structures are depicted in a different color to the appended substituents, the authors should ensure this is consistent throughout, e.g. structure of pyrotinib.

Response:

The caption of Fig. 3 was changed.

Regarding the EGR-TKIs in Fig. 3, their chemical structure were revised. We kept the basic heterocyclic nucleus in the black color. We tried to keep a uniform color through the whole manuscript, however in limited cases different colors were given to certain groups such as in the synthesis/metabolism in order to make it easy for the reader to follow.  

The figures of pyrotinib (57-60) were changed to match with Fig. 3.

Comment:

  1. Lines 93-94: Change to "….. the primary focus is on small-molecule inhibitors of EGFR which are approved for clinical use as anticancer agents."

Response: it was changed in the revised manuscript

Comment:

  1. Line 99: Change to "Based on their chemical structures, the approved EGFR-TKIs can be …."

Response: it was changed in the revised manuscript

Comment:

  1. Line 101: Change to "…., olmutinib is considered….."

Response: it was corrected in the revised manuscript

Comment:

  1. Figure 4: For the quinazoline-4,6-diamine subtype, the 6-amino group should be in blue color.

Response: It was changed in the revised manuscript

Comment:

  1. Line 110: Change to "can also be classified as reversible or irreversible inhibitors (Fig. 5)."

Response: It was changed in the revised manuscript

Comment:

  1. Line 110-112: Change to something like "For the first type, the inhibitors competitively bind to the ATP binding site in EGFR through noncovalent interactions involving electrostatic, hydrogen-bonding, and hydrophobic interactions"

Response: It was corrected in the revised manuscript

Comment:

  1. Lines 114-116: Is it true that these irreversible inhibitors always react with the cysteine residues in EGFR? In any event, change the last sentence to something like "The structure of these inhibitors is characterized by the presence of an electrophilic side chain that acts as a Michael acceptor which chemically reacts with the cysteine thiol group to form the addition product."

Response:

The covalent interaction with Cys residue was reported in the majority of the cited references. However, we observed that at least one of these inhibitors, osimertinib was reported in 3 crystal structures forming only non-covalent interaction.

It was changed in the revised manuscript.

Comment:

  1. Figure 5: Change caption to "Classification of EGFR-TKIs based on nature of inhibition of EGFR".

Response: It was changed in the revised manuscript

Comment:

  1. Lines 121-123: Change to "The EGFR inhibitors can also be classified into three groups based on the treatment of the types of cancers for which they are approved [26]. The first group includes the drugs which are approved for the treatment of NSCLC such as afatinib….." <Note that the acronym NSCLC must be defined where it first appears - non-small-cell lung carcinoma>.

Response:

It was changed in the revised manuscript

NSCLC was defined where it was first mentioned “line 80”.

Comment:

  1. Line 144: What is the advantage of the second-generation EGFR-TKIs over the first generation EGFR-TKIs?

Response: The first-generation EGFR-TKIs lack the Michael acceptor site. accordingly, the act as competitive/reversible inhibitors of EGFR, while the second have the Michael acceptor site which allow them to irreversibly inhibitor EGFR. The paragraph describing the difference and advantaged in the MS was revised and updated to point out this difference.

Comment:

  1. Lines 150-151: Change to something like "These moieties bear an acrylamide group that as noted above forms a covalent bond with the cysteine residue in EGFR" <use of moiety in this specific context is not correct; group is much better>

Response: It was changed as suggested in the revised manuscript. 

Comment:

  1. Lines 156-159: Change to something like "The rapid emergence of EGFR T790M and C797S mutations has led to the development of resistance to the third-generation EGFR-TKIs [30]. <the authors may like to note the C797S mutation deprives the ability of the 3rd gen TKIs to act as irreversible inhibitors, as the Michael donor group – the SH in cysteine is replaced by the ineffective –OH in serine>. In addition, the emergence of EGFR double and triple (del19/L858R + C797S ± T790M) <this should be expanded upon and explained more carefully> mutations challenges the therapeutic effectiveness of EGFR-TKIs. These problems underscore the ongoing need to develop new and potent EGFR-TKIs."

Response: It was changed as suggested in the revised manuscript. 

Comment:

  1. Lines 161-164: the development of the allosteric EGFR inhibitors is a noteworthy achievement, although it is recognized that it would not be easy to incorporate structural aspects into the current review; change to "Recently, several fourth generation allosteric EGFR inhibitors that bind to a site in EGFR other than the PTK domain were reported [32]. However, although these inhibitors are ineffective against NSCLC with mutated EGFR, they display synergistic anticancer effects when combined with the current EGFR inhibitors" <<will help to indicate which ones>>

Response: It was changed as suggested in the revised manuscript

Comment:

  1. Lines 165-166: best to combine this paragraph with the preceding paragraph and as structures for EAI045, BLU-945, and BBT-176 are not given, perhaps best to change to something like "Currently, several fourth generation EGFR-TKIs are being evaluated for efficacy against cancers carrying double or triple EGFR mutations [33–35]. One of these compounds, BBT-176 is in Phase I/II trials in NSCLC patients with advanced lung cancer [35], but these inhibitors are not yet approved for clinical use." Otherwise insert a figure showing structures for these fourth generations TKIs and leave descriptors in the text.

Response: It was changed as suggested in the revised manuscript

Comment:

  1. The next part of the review comprises descriptions of the individual EGFR inhibitors, with overviews of the approval history, synthesis, target kinases, details of the inhibitor binding, biological activity, and metabolism. The presentation of such data overall represents the main strength of this review and is richly informative. However, the authors should amend and improve these sections according to the foregoing.

Response: We would like to thank the reviewer for his positive comments

Comment:

  1. The Schemes overall need to be visually enhanced; at the moment the diagrams and reaction arrows are not so attractive, the captions should be consistent, e.g. Scheme 1. Synthesis of afatinib, and in certain cases there are mistakes.

Response:

All schemes were updated to enhance their resolutions.

The caption of all schemes were revised and corrected. 

Comment:

  1. Scheme 1: hashed bond in compound 4 and subsequent needs to be displayed properly.

Response: Scheme 1 was updated, the hashed bonds were displayed clearly.

Comment:

  1. Lines 195-196: Change to "Afatinib inhibits ……..In addition, it shows …" <here and in numerous places in the following text when describing efficacies as reflected in IC50 and other data>

Response: It was changed corrected in the revised manuscript

Comment:

  1. Line 211 and Figures 9, 10: Did the authors use Discover Studio Visualizer to generate the binding modes in Figures 9 and 10?; if so, this should be stated in the caption to this, and ALL subsequent figures where this program is used.

Response: Discover Studio Visualizer was cited in all the figures, where it was used.

Comment:

  1. Lines 233-234: Change to "The formation of this covalent bond leads to irreversible inhibition of the autophosphorylation of the ErbB family [41]. The mechanism of the irreversible inhibition of EGFR by afatinib is illustrated in Fig. 11."

Response: It was changed revised and corrected in the revised manuscript

Comment:

  1. Lines 238-242: Change to something like "Afatinib binds to and inhibits EGFR with exon 19 deletion mutations and exon 21 L858R mutations. The covalent interaction leads to irreversible inhibition of the kinase activity which provides an advantage over other non-covalent competitive inhibitors of EGFR such as erlotinib and gefitinib [43–45]."

Response: It was changed revised and corrected per the reviewer’s suggestion.

Comment:

  1. Line 258 Metabolism - here and for all subsequent compounds these sections have to be improved, particularly with regard to the Figures illustrating the metabolic conversions. Thus in Figure 12 the caption must be changed from "The main metabolic of afatinib must be changed to something like "the principal metabolic pathway for afatinib". The Figure 12 is poorly presented: the reaction indicating addition of 'RH' has to be changed to 'RSH' and the curved arrows changed to indicate the correct chemical sense (cf. Figure 11 displaying addition of Cys-797); also one presumes that the authors know the reaction is reversible. The statement in lines 282-284 tends to indicate this is true.  Accuracy is important for ensuring credibility of a review.

Response:

We think that using the term “the proposed metabolic pathways” is more accurate like what was mentioned in the captions of the corresponding figure in the cited references.

Figure 12 was updated and corrected in the revised manuscript.

We also revised the cited reference to confirm that “a covalent protein adducts with afatinib” is formed. These words (between the above brackets) describing the nature of the interaction was taken as copy/paste from the cited reference.

The statement in lines 282-284 was changed using the same words in the cited reference “ The slow decomposition of this adduct could lead to prolongation of the elimination phase”. 

Comment:

  1. Lines 277-284: There are spelling mistakes that must be corrected.

Response: It was changed revised and corrected in the revised manuscript

Comment:

  1. For the succeeding sections, the following are briefly noted.
  1. line 316, "The crystal structure of almonertinib bound to any of its target kinases has not yet been reported"; ii. structures in Figure 15 are incomprehensible even though a brief explanation is given below (lines 334-339); compare for example Figure 20, where structures are clearly shown iii. structures in Scheme 3 are not color-coded as in the preceding schemes, and others.

Response:

Line 316: was revised and corrected in the revised manuscript.

Figure 15: the names of the metabolic pathways were added and the figure was updated to make it easier for the reader to expect the metabolite at each pathway.

Scheme 3 was updated with color-coded structure

  1. The authors should use the foregoing to improve the manuscript overall.

Response:

The whole manuscript was revised for typo/grammar mistakes. The syntax of many sentences was modified. Schemes and metabolism figures were revised and updated to improve the resolution. 

Comment:

Finally, to help the authors, the Conclusions and Perspective may be written as follows – one does trust the meaning has not been changed from what the authors originally intended:

  1. Conclusion

Targeting the tyrosine kinase domain in wild-type and mutants of EGFR with small-molecule inhibitors is confirmed as a valid strategy in cancer chemotherapy.  Since the approval of the first EGFR-TKI, erlotinib, considerable effort has been devoted to the discovery and development of new potent and safe inhibitors. Fourteen EGFR small-molecule inhibitors have so far been approved for the treatment of different types of cancers. The primary focus of this review is on EGFR-TKIs which have been approved for the treatment of different types of cancers. These drugs are classified based on their chemical structures, target kinases, and pharmacological uses. In addition, the synthetic routes to each of these drugs are discussed.  The binding modes and interactions of these drugs into their target kinases have been visualized and discussed. Based on the nature of the binding interactions with the target kinases, these drugs may be classified as reversible or irreversible inhibitors. The cytotoxic activities of these drugs against different types of cancer cell lines have also been summarized. In addition, the metabolic pathways and the various metabolites associated with the fourteen drugs are also presented. However, the effectiveness of these drugs is challenged by the development of single, double, and triple mutations in EGFR. Recent results from preclinical and clinical studies of fourth generation EGFR-TKIs indicate these may provide effective treatment for patients with such mutations. Overall, this review highlights the syntheses, target kinases, crystal structures, binding interactions, cytotoxicity, and metabolism of the fourteen approved EGFR inhibitors. These data should greatly help in the design of new EGFR inhibitors. 

  1. Perspective

Currently, first- to third-generation EGFR-TKIs are in clinical use for the treatment of different types of cancers. Although high treatment efficacy is established, the development of several muations within EGFR [165] affect drug response rates in NSCLC patients [166]. The EGFR C797S mutation has rapidly led to resistance to the third generation EGFR-TKIs [33], such that these are becoming ineffective in patients with these mutations [32]. Accordingly, the primary focus of the research in this area is directed toward the discovery and development of new therapeutics that can target EGFR carrying the del19/L858R+ C797S ± T790M mutations. Recently, fourth generation allosteric mutant-selective EGFR inhibitors have been developed that show synergistic anticancer effects against NSCLC with mutant EGFR when combined with traditional EGFR inhibitors [32]. In addition, several fourth generation EGFR-TKIs (EAI001, EAI045, BLU-945 and BBT-176) are being investigated against cancer cells with double or triple EGFR mutation [32–35]. One of these compounds, BBT-176 is in Phase I/II trial in NSCLC patient with advanced lung cancer [35]. Should one of these drugs at least reach the market, this will greatly encourage further research in this area.

Response:

We would like to express our deep thanks to reviewer 1 for his kind help and accurate revision which really improved the quality of the MS. The suggested text of conclusion/perspective will help too much in improving the quality of this manuscript. The text was revised and added to the final manuscript with few modifications.

Reviewer 2 Report

The quality of the manuscript was improved significantly. 

Just a few further indications: line 722, structures M6 and M7 are in fact regioisomers, not geometric isomers. I apologize to have missed this indication before.

Line 748-749:  indicate that compound 67 was obtained by coupling 65 with 66 and by reducing the nitro group to amino by Fe and acids (two steps)!

Author Response

Reviewer 2

Comments and Suggestions for Authors

The quality of the manuscript was improved significantly. 

Response: We thank the reviewer for this positive comment.

Comment: Just a few further indications: line 722, structures M6 and M7 are in fact regioisomers, not geometric isomers. I apologize to have missed this indication before.

Response: Geometric isomers was changed to “regioisomers”

Comment: Line 748-749:  indicate that compound 67 was obtained by coupling 65 with 66 and by reducing the nitro group to amino by Fe and acids (two steps)!

Response: it was corrected in the revised manuscript.

Reviewer 3 Report

This is a thorough review on EGFR inhibitors. I would suggest to add reaction yields in the schemes of inhibitors synthesis

Author Response

Reviewer 3

Comments and Suggestions for Authors

This is a thorough review on EGFR inhibitors. I would suggest to add reaction yields in the schemes of inhibitors synthesis

Response: We highly appreciate the valuable comments of reviewer 3 and his valuable corrections that have been emerged after his careful and precise revision which would help in improving the quality of the manuscript.

The yield% was added to all steps in the 14 schemes. However, some steps in which the original reference did not mention the yield%, or used a qualitative term to describe the yield, or mentioned the yield in weight units. In such cases, the yield% was not added.     

Round 2

Reviewer 1 Report

MS is improved, and is evolving into a useful review.  However, as previously noted, the authors MUST carefully check ALL figures and schemes, especially also to ensure that structures and transformations are correct, and that the presentations are consistent. Unfortunately, there are still  too many obscure aspects to the figures and schemes, too many acronyms, that should be explained in the figure and scheme captions, which are too brief. In general also, the metabolism figures still are inconsistently presented; in addition the use of an asterisk (*) to designate position of label is not so informative in the absence of any proper description; as it does not add information, may be better to leave the asterisk out.  Also brief labelling over the arrows for the types of metabolism involved would be helpful.  Thus, to assist the authors, a few more examples are now presented.

              Line 189 and Scheme 1, structure of intermediates leading to product are wrong cf. Fig. 7 - tetrahydrofuran-3-ol was used, and Scheme must be redrawn correctly. Likewise in Fig. 11, structure of afatinib with respect to the tetrahydrofuranyl unit is incorrect.  In Scheme 2, also Scheme 12, no. of significant figures for the yields is not sensible – best to round up or down to nearest integer (although for Scheme 2, this regrettably appears to be copied from ref. 52, which itself is a review, largely of the syntheses of the tyrosine kinase inhibitors; authors should cite original reference). Indole is indole, and no need to designate "1H". Figure 15, the metabolic pathways remain difficult to comprehend -  e.g. the unpleasant word 'dedimethylethylaminization' presumably refers to oxidative dealkylation of the 2'-(N,N-dimethylamino)ethyl group, so why is there a hashed blue rectangle drawn around this group which is retained in M511b and M632b?  Michael addition of a cysteine residue to the acrylamide will give an adduct comprising the bound cysteine residue and "1H"- what does the "2H" mean?  Why not draw out the actual –C(=O)CH2CH2-Cys with a single bond and remove the hashed red rectangle in all relevant structures? (also for other figures, e.g. Figure 56 for Glu in M18, etc.). It is also not clear to suggest that acetylation of cysteine takes place after the Michael addition; surely it is N-acetylcysteine which adds, as noted in the text. In M632b, what does the pendant –CH2 signify?  In Figure 20, whilst it is appreciated the internal piperidine group attached to the aromatic ring of brigatinib has a basic nitrogen that will be protonated at physiological pH, hydroxylation via N-oxide formation and then intercession of the Polonovski reaction via formation of the iminium ion to yield the a-hydroxyl morpholino products are incorrectly depicted as protonated species, especially for the α-keto-α'-hydroxyl product drawn below brigatinib, which assuredly will not be protonated, and certainly not on the hydroxyl group. This also applies to the iminium adducts BGB623 and BGB527 trapped by cyanide. As the types of experiments carried out by Kadi involve use of cyanide, these nitrile-substituted end products are not metabolism products as such. This should be briefly commented upon in the scheme caption.  This also applies to the use of cyanide to intercept iminium products during the metabolism of dacomitinib (Figure 24).  In Scheme 4, T3P is presumably Ph3P; this should be written as such. If so, this is presumably a redox type condensation between the carboxylic acid 29 and aromatic amine 28, and therefore, a reagent is missing – is iodine used or is the reaction carried out in CCl4? MeCN is incorrectly depicted.  In Figure 24, why is the "cyano-addition" (which has nothing to do with metabolism) inserted before the piperidine unit in dacomitinib, whereas no such designation appears in Figure 20? In Scheme 5, what is TBAI – presumably tetra-n-butylammonium iodide? (compare Scheme 13 where tetra-butyl ammonium chloride is written out in full). In Figure 29, presumably M3 is the glucuronidation product? – the use of the formula – OC6H8O6H – does not properly inform.  In Scheme 6, what is i-ProOH?   In Scheme 8, what is Cy3P? In figure 42, what is M5 and how is it formed from M2? – presumably, there has to be a positive charge on N bearing the -CH2CH2SO2CH3 (?) group.  In Figure 47, what does the –O mean attached to the hashed rectangles in M4, M5 and M8?; write the N-oxide M12 in the usual manner R3N+-O(cf. M1 in Figure 56, N-oxide in Figure 65. and N-oxidation in Figure 66); presumably –SG implies glutathionylated adducts in M1, M2, M4-M6 (cf. M9 and M11); whilst briefly mentioned in the text, this could be mentioned in the caption; compare Figure 50, where GSH is written over the arrow leading to gluthathionylation of the quinone imine. In Scheme 11, what is DMAC? (cf. DMA reagent leading to 83), and DMA in Scheme 13, where DMA is mentioned as dimethyl acetal for the first time). In Figure 65, -Glu is taken to be the glucuronide conjugate, but elsewhere, the acronym implies glutamate.  Captions of all figures should be checked for correct grammatical constructs.

Finally brief examples of grammatical aspects, although should not be mentioned as part of the reviewing process. As before these are not complete, and expressions must be improved.

  1. lines 46-49, best in present tense something like: "The overexpression of EGFR is associated with different types of cancers [3,4]. Accordingly, targeting EGFR with small molecules inhibitors is confirmed as a valid strategy in cancer therapy [5]. This area has attracted the researchers during the last two decades where large number of EGFR small-molecule inhibitors have been developed [6–8]."
  2. Lines 59-60: best as something like: "The kinase domain (Fig. 2) consists of two lobes designated the N- and C-lobes respectively [12]."
  3. Line 85, delete 'in' before [23], and change to "However, during the second decade…"
  4. Line 144, change to "….. of the kinase activity that provides…"
  5. Line 330, change to "Representative pathways to these metabolites are presented in Fig. 15"
  6. Line 431-432, and elsewhere, as noted previously, watch use of tense – thus change to "The binding mode and interactions of dacomitinib into EGFR T790M kinase domain 431 (pdb: 4I24) is visualized in Fig. 23
  7. Line 597, change to "…of gefitinib were identified (Fig. 34)."
  8. Line 600-601, change to "Wang et al. investigated the metabolism of gefitinib in NSCLC patients [92]. Eighteen metabolites were tentatively identified in human plasma."  

Author Response

Comments and Suggestions for Authors

MS is improved and is evolving into a useful review.  However, as previously noted, the authors MUST carefully check ALL figures and schemes, especially also to ensure that structures and transformations are correct and that the presentations are consistent. Unfortunately, there are still too many obscure aspects to the figures and schemes, too many acronyms, that should be explained in the figure and scheme captions, which are too brief. In general also, the metabolism figures still are inconsistently presented; in addition the use of an asterisk (*) to designate position of label is not so informative in the absence of any proper description; as it does not add information, maybe better to leave the asterisk out.  Also, brief labeling over the arrows for the types of metabolism involved would be helpful.  Thus, to assist the authors, a few more examples are now presented.

Response: We highly appreciate the reviewers’ valuable comments and corrections that have been emerged after their careful and precise revision, which would help in improving the quality of the manuscript.

Acronyms were defined in the text. In some cases, the full name was only used. The asterisks * indicating the position of the radiolabel were removed. The metabolic pathways were added in most of the metabolism figures (based on the cited references). We indicated the revisions/corrections by a cyan highlighter in the revised manuscript. Below, are our responses to the comments, point-by-point.

Comment: Line 189 and Scheme 1, the structure of intermediates leading to the product are wrong cf. Fig. 7 - tetrahydrofuran-3-ol was used, and Scheme must be redrawn correctly.

Response: We thank reviewer 1 for this correction. The chemical structures of the compounds in scheme 1 were revised and corrected.

Comment: Likewise in Fig. 11, the structure of afatinib with respect to the tetrahydrofuranyl unit is incorrect. 

We thank reviewer 1 for this correction. Fig. 11 was revised and corrected

Comment: In Scheme 2, also Scheme 12, no. of significant figures for the yields is not sensible – best to round up or down to nearest integer (although for Scheme 2, this regrettably appears to be copied from ref. 52, which itself is a review, largely of the syntheses of the tyrosine kinase inhibitors; authors should cite the original reference).

Response: The values of the yield% in schemes 2 and 12 were rounded up to the nearest integer number.

Comment: Indole is indole, and no need to designate "1H".

Response: It was corrected

Comment: Figure 15, the metabolic pathways remain difficult to comprehend -  e.g. the unpleasant word 'dedimethylethylaminization' presumably refers to oxidative dealkylation of the 2'-(N,N-dimethylamino)ethyl group, so why is there a hashed blue rectangle drawn around this group which is retained in M511b and M632b? 

Response: The “dedimethylethylaminization” was replaced by oxidative dealkylation of the 2'-(N,N-dimethylamino)ethyl group

Comment: Michael addition of a cysteine residue to the acrylamide will give an adduct comprising the bound cysteine residue and "1H"- what does the "2H" mean?  Why not draw out the actual –C(=O)CH2CH2-Cys with a single bond and remove the hashed red rectangle in all relevant structures? (also for other figures, e.g. Figure 56 for Glu in M18, etc.).

Response: We agree with this suggestion. Fig. 15 was revised and updated. The hashed square around the metabolic sites was removed. The figure was also updated. The metabolic pathways were added over the arrows. The acronym “GU” was used for glucuronic acid conjugate. 

In Fig. 56 we thank that it is better to leave the structure as it is because the functional group which can bind to glucuronic acid was not identified.

Comment: It is also not clear to suggest that acetylation of cysteine takes place after the Michael addition; surely it is N-acetylcysteine which adds, as noted in the text.

Response: We agree with the reviewer. Although the original article presents the two metabolites sequentially, with an arrow indicating that the cysteine conjugate gives the acetylcysteine conjugate.

Fig 15 was updated per the reviewer’s suggestion

Comment: In M632b, what does the pendant –CH2 signify? 

Response: Although it was not described in the original reference (ref No 56), the (▬ - CH2) was found in the metabolites which undergo demethylation.

Since the demethylation process involves the removal of one methyl group (X-CH3, X = N, N, S,), leaving one H atom attached to the heteroatom. The overall change in the chemical structure will be the removal of CH2 group. The authors put a negative sign before the (▬ - CH2) to indicate the removal of this group.

Comment: In Figure 20, whilst it is appreciated the internal piperidine group attached to the aromatic ring of brigatinib has basic nitrogen that will be protonated at physiological pH, hydroxylation via N-oxide formation, and then intercession of the Polonovski reaction via formation of the iminium ion to yield the a-hydroxyl morpholino products are incorrectly depicted as protonated species, especially for the α-keto-α'-hydroxyl product drawn below brigatinib, which assuredly will not be protonated, and certainly not on the hydroxyl group. This also applies to the iminium adducts BGB623 and BGB527 trapped by cyanide. As the types of experiments carried out by Kadi involve the use of cyanide, these nitrile-substituted end products are not metabolism products as such. This should be briefly commented upon in the scheme caption. This also applies to the use of cyanide to intercept iminium products during the metabolism of dacomitinib (Figure 24).  

Response: We agree with this comment. A sentence was added to the legend of Fig. 20  to indicate that the three nitrile derivatives (BGB609, BGB623, and BGB527) are not metabolic products of brigatinib. 

The legend in Fig. 24 was also updated like Fig. 20.

Comment: In Scheme 4, T3P is presumably Ph3P; this should be written as such. If so, this is presumably a redox type condensation between the carboxylic acid 29 and aromatic amine 28, and therefore, a reagent is missing – is iodine used or is the reaction carried out in CCl4?

Response: T3P, 1-propanephosphonic acid cyclic is used with 2,6-lutidine in the amide bond formation. The acronym and role of this reagent were added in the text in the revised MS. 

Comment: MeCN is incorrectly depicted. 

Response: The acronym of acetonitrile was kept as it is in scheme 4.

Comment: In Figure 24, why is the "cyano-addition" (which has nothing to do with metabolism) inserted before the piperidine unit in dacomitinib, whereas no such designation appears in Figure 20?

Response: Fig. 24 was revised. The hashed square and "cyano-addition" designation were removed.

Comment: In Scheme 5, what is TBAI – presumably tetra-n-butylammonium iodide? (compare Scheme 13 where tetra-butyl ammonium chloride is written out in full).

Response: Scheme 5 was revised and updated. TBAI, Tetrabutylammonium iodide was used as full name.

Comment: In Figure 29, presumably M3 is the glucuronidation product? – the use of the formula – OC6H8O6H – does not properly inform. 

Response: Fig. 29 was updated. The indicated formula of glucuronic acid was replaced by GU. The metabolic pathways were added in fig. 29 to make it clear for the readers.

Comment: In Scheme 6, what is i-ProOH?  

Response: Isopropyl alcohol/isopropanol, the acronym was defined in the text in the revised manuscript.

Comment: In Scheme 8, what is Cy3P?

Response: Cy3P, Tricyclohexylphosphine tetrafluoroborate, a ligand used with Pd(OAc)2 in the Pd-catalyzed arylation of compound 61. Acronym and definition were added in the revised MS. 

Comment: In figure 42, what is M5 and how is it formed from M2? – presumably, there has to be a positive charge on N bearing the -CH2CH2SO2CH3 (?) group.

Response: We thank the reviewer for this correction. Figure 42 was revised and corrected. The missing charge was added. Following the bioactivation of the furane ring, an intramolecular cyclization involving the secondary amine takes place leading to the formation of pyridinium salt M5. hydroxypyridine metabolite M2 was formed on the loss of the ethyl sulfone moiety from M5. A brief description of the two metabolites (M2 and M5) was included in the text in the revised manuscript.

Comment: In Figure 47, what does the –O means attached to the hashed rectangles in M4, M5 and M8?;

Response: It means oxygenation (oxidation) of the picoline moiety. The hashed rectangles indicate that oxygenation can take place at different positions of the picoline moiety.

Comment: Write the N-oxide M12 in the usual manner R3N+-O‑ (cf. M1 in Figure 56, N-oxide in Figure 65. and N-oxidation in Figure 66);

Response: The N-oxide was corrected in Fig. 47.

Comment: Presumably –SG implies glutathionylated adducts in M1, M2, M4-M6 (cf. M9 and M11); whilst briefly mentioned in the text, this could be mentioned in the caption; compare Figure 50, where GSH is written over the arrow leading to gluthathionylation of the quinone imine.

Response: Fig. 47 was updated. The metabolic pathways including GSH are written over the.  

Comment: In Scheme 11, what is DMAC? (cf. DMA reagent leading to 83), and DMA in Scheme 13, where DMA is mentioned as dimethyl acetal for the first time)..

Response:

The acronym: DMAC, N,N-dimethylacetamide. DMF–DMA, N,N-Dimethylformamide dimethyl acetal.

DIPEA diisopropyl ethylamine , DCM, dichloromethane. All were defined in the text or mentioned as full names without the acronym.

Comment: In Figure 65, -Glu is taken to be the glucuronide conjugate, but elsewhere, the acronym implies glutamate.  Captions of all figures should be checked for correct grammatical constructs

Response: The acronym of glucuronide acid (GU) was used in all metabolism figures. In addition, the names of the metabolites in Fig. 65 were added to. Captions of all figures were checked for any grammatical constructs

Comment: Finally brief examples of grammatical aspects, although should not be mentioned as part of the reviewing process. As before these are not complete, and expressions must be improved.

lines 46-49, best in the present tense something like: "The overexpression of EGFR is associated with different types of cancers [3,4]. Accordingly, targeting EGFR with small molecules inhibitors is confirmed as a valid strategy in cancer therapy [5]. This area has attracted researchers during the last two decades where large number of EGFR small-molecule inhibitors have been developed [6–8]."

Response: It was corrected in the revised manuscript.

Comment: Lines 59-60: best as something like: "The kinase domain (Fig. 2) consists of two lobes designated the N- and C-lobes respectively [12]."

Response: It was corrected in the revised manuscript.

Comment: Line 85, delete 'in' before [23], and change to "However, during the second decade…"

Response: It was corrected in the revised manuscript.

Comment: Line 144, change to "….. of the kinase activity that provides…"

Response: It was corrected in the revised manuscript.

Comment: Line 330, change to "Representative pathways to these metabolites are presented in Fig. 15"

Response: It was corrected in the revised manuscript.

Comment: Line 431-432, and elsewhere, as noted previously, watch use of tense – thus change to "The binding mode and interactions of dacomitinib into EGFR T790M kinase domain 431 (pdb: 4I24) is visualized in Fig. 23

Response: This sentence was revised and corrected.  Similar sentences in the manuscript were also revised.

Comment: Line 597, change to "…of gefitinib were identified (Fig. 34)."

Response: The syntax o this sentence was changed in the revised manuscript.

Comment: Line 600-601, change to "Wang et al. investigated the metabolism of gefitinib in NSCLC patients [92]. Eighteen metabolites were tentatively identified in human plasma."  

Response: We thank reviewer 1 for this comment. The two sentences were changed as suggested.

Reviewer 2 Report

On the whole, the paper can be considered a comprehensive and thorough account of the chemistry and biological activity of a class of kinases inhibitors employed as anticancer agents in therapy.

The authors report a good deal of data and the structure of their presentation has been well designed.

However, a number of important concerns arise regarding both chemical concepts and the Language. For the latter, I strongly suggest the intervention of a native British speaker or a professional editing service.

Examples of the mistakes and the unclear points are given below:

Line 56,57: the sentence is not clear;

Line 64: the use of the past tense is not correct. Please, write the verbs in the present simple form: “….analogue of ATP are visualized in fig. 2…..exhibits two important…..”.

Line 70: “visualized” as for line 64.

Line 74: change “researcher” with “researchers” (plural).

Line 75: writw: “a large number”

Line 85: delete “in”

Line 85: write “The first one….”

Line 91: write “…that were approved…”

Line 99: “approved” is written two times.

Line 111: delete “competitive” and write “…as a competitor..”.

Line 116,117: whit does “…an addition complex” means?

Line 165: the identification codes of the fourth generation inhibitors do not say anything; please add also the chemical structures.

Line 192, 1923: delete “afforded”. Further , the reaction of 8 with 9 (NOT with 7) gives Afatinib!

Line 211: as for line 64.

Line 215: delete “amino acids”, it is pleonastic.

Line 235: as for 64.

Line 240-242: please, rewrite the sentence as it is unclear.

Line 259: “In an attempt…” does not match with “several studies” (singular with plural).

Line 268, 269: change “faces” with “feces”

Figure 12: “R” cannot represent a heteroatom.

Line 296: change “..compound 13” with “…compound 14”.

Line 328: “metabolism” is written twice.

Line 329: Is a rat a “Chinese subject”

Line 356: Molecular hydrogen is the main reactant to obtain 23; Pd/C is just the catalyst. Further, 23 and 26 are coupled by a Pd catalysed cross-coupling in which potassium carbonate is just a basicity adjuster. Rewrite all the description.

Line 372 “amino acids” is pleonastic.

Line 395-397: the meaning is not understandable.

Line 401: What does “to undergo a metabolic pathway” exactly mean?

Scheme 4: what is “MeCn”?

Line 442: is it sure the dose is expressed in grams?

Line 500: as for 64.

Line 534: write “…approved for..”

Line 541: as for 64.

Line 570, figure 32 and many other points afterward: what do “carbon H bond” and “covent H bond” exactly mean in the context of the ligand-receptor interactions? I never heard of them. I cannot find them in any of the cited references. Please, clarify better or tell me something more. Thank you.

Scheme 7: change the “z” with the “O”.

Line 628: I cannot see the utility of HCl. Can you explain better?

Line 648: Among must be written in capital initial letter.

Line 695: as for 64.

Line 702: use the present simple, not the present continuous.

Line 705: write “...revealed a growth…

Figure 42: the structure M7 is uncorrect. There is a CH2 more.

Line 721 and 722: geometric isomers and stereoisomeric isomers (used for the oximes) are exactly the same. Please use a uniform notation.

Line 727-728: there no reference.

Scheme 9: compound 65 is a nitro-derivative that is reduced by Fe and acid. It is not an amino-derivative (see ref. 117). Further, the structure of 66 must be checked. I guess it is uncorrect.

Line 764: the sentence lacks the verb!

Line 781: Also here lacks the verb.

Line 810: write “…treatment of patients…”

Line 831: “also” is not correctly placed.

Scheme 11: please, check the structures of compounds 82 and 83.

Line 887: what does “cellular potency” exactly mean?

Line 899: as for 64.

Line 905: as for 64.

Line 935: write “feces”, NOT “faces”

Line 955 write “were” (plural) instead of “was” (singular).

Line 986: “healthy males” of which animal?

Line 999: It is obvious that a “EGFR-TKI” acts by inhibition of a kinase

Line 1042: write better the sentence. Maybe: “a half-life consistent with that of …”.

Line 1051-1052: the sentence is redundant. Read it carefully!

Scheme 13: What is DMA? It is not clear.

Line 1059: Is 96 a “nitro group”? maybe a nitro compound.

Line 1064: I suppose it is not 88, but 99.

Line 1070: Why the word “mechanistic”

Line 1095: the cell line MDA-MB-231 is repeated three times!!

Line 1100: change “studies” with “studied”.

Line 1127: use the present perfect, NOT the past simple.

Line 1142: change “are being” with “have been”.

Line 1143-1146: is a repetition. Please delete it.

Line 1160: change “are being” with “have been”.

In conclusion, it is my opinion that the manuscript need to be partially rewritten and then carefully checked by reading it several times before to submit it further.

Author Response

Reviewer 2

Comments and Suggestions for Authors

On the whole, the paper can be considered a comprehensive and thorough account of the chemistry and biological activity of a class of kinases inhibitors employed as anticancer agents in therapy. The authors report a good deal of data, and the structure of their presentation has been well designed. However, a number of important concerns arise regarding both chemical concepts and the Language. For the latter, I strongly suggest the intervention of a native British speaker or a professional editing service.

Examples of the mistakes and the unclear points are given below:

We highly appreciate the valuable comments of reviewer 2 and his valuable corrections that have been emerged after his careful and precise revision which would help in improving the quality of the manuscript. We have revised the manuscript 3 times. Definitions were added where acronyms first appear. Spelling and typographical errors, errors of grammar such as the use of wrong tense in sentences, or wrong descriptors or incorrect definitions were also revised and corrected. The schemes and figures were also revised and corrected. All the revisions/corrections done in the revised manuscript were highlighted with a yellow highlighter. Below, are our responses to the comments, point-by point.

Comment: Line 56,57: the sentence is not clear.

Response: The syntax of this sentence was changed in the revised manuscript. 

Comment: Line 64: the use of the past tense is not correct. Please, write the verbs in the present simple form: “….analogue of ATP are visualized in fig. 2…..exhibits two important…..”.

Response: The tense of the verbs was changed in this sentence and in similar sentences throughout the whole manuscript

Comment: Line 70: “visualized” as for line 64.

Response: It was corrected in the revised manuscript

Comment: Line 74: change “researcher” with “researchers” (plural).

Response: It was corrected in the revised manuscript

Comment: Line 75: writw: “a large number”

Response: It was corrected in the revised manuscript

Comment: Line 85: delete “in”

Response: It was corrected in the revised manuscript

Line 85: write “The first one….”

Response: it was corrected in the revised manuscript.

Comment: Line 91: write “…that were approved…”

Response: it was corrected in the revised manuscript.

Comment: Line 99: “approved” is written two times.

Response: it was corrected in the revised manuscript.

Comment: Line 111: delete “competitive” and write “…as a competitor..”.

Response: the syntax of the sentence was changed to give a similar meaning.  

Comment: Line 116,117: whit does “…an addition complex” means?

Response: it was replaced by “a covalent adduct” in the revised manuscript.

Comment: Line 165: the identification codes of the fourth generation inhibitors do not say anything; please add also the chemical structures.

Response: Because it was not easy to incorporate the chemical aspects of the fourth-generation EGFR inhibitors into the current review, where we aimed to focus only on the inhibitors approved for clinical use, and to avoid any shift in figure numbers, we have modified the syntax of this part in the revised manuscript.

Comment: Line 192, 1923: delete “afforded”. Further , the reaction of 8 with 9 (NOT with 7) gives Afatinib!

Response: it was corrected in the revised manuscript.

Comment: Line 211: as for line 64.

Response: it was corrected in the revised manuscript.

Comment: Line 215: delete “amino acids”, it is pleonastic

Response: it was deleted from the revised manuscript.

Comment: Line 235: as for 64.

Response: it was corrected in the revised manuscript.

Comment: Line 240-242: please, rewrite the sentence as it is unclear.

Response: The syntax was revised and rewritten in the revised manuscript.

Comment: Line 259: “In an attempt…” does not match with “several studies” (singular with plural).

Response: The syntax was revised and corrected

Comment: Line 268, 269: change “faces” with “feces”

Response: lines 268 and 269 were corrected. 

Comment: Figure 12: “R” cannot represent a heteroatom.

Response: FIG. 12 was corrected in the revised manuscript.

Comment: Line 296: change “..compound 13” with “…compound 14”.

Response: It was corrected in the revised manuscript

Comment: Line 328: “metabolism” is written twice.

Response: It was corrected in the revised manuscript

Comment: Line 329: Is a rat a “Chinese subject”

Response: It was corrected in the revised manuscript

Comment: Line 356: Molecular hydrogen is the main reactant to obtain 23; Pd/C is just the catalyst. Further, 23 and 26 are coupled by a Pd catalyzed cross-coupling in which potassium carbonate is just a basicity adjuster. Rewrite all the description.

Response: The two sentences were rewritten in the revised manuscript.

Comment: Line 372 “amino acids” is pleonastic.

Response: It was corrected in the revised manuscript.

Comment: Line 395-397: the meaning is not understandable.

Response: The syntax of this part was revised and modified in the revised manuscript.

Comment: Line 401: What does “to undergo a metabolic pathway” exactly mean?

Response: This sentence was revised and corrected in the revised manuscript.

Comment: Scheme 4: what is “MeCn”?

Response: MeCN is an abbreviation of Acetonitrile. It was also used in the cited reference No 64 (J. Med. Chem. )

Comment: Line 442: is it sure the dose is expressed in grams?

Response: The text in lines 440-443 was revised. The syntax was modified to avoid this term (IC50 g).

Comment: Line 500: as for 64.

Response: It was corrected in the revised manuscript.

Comment: Line 534: write “…approved for..”

Response: It was corrected in the revised manuscript.

Comment: Line 541: as for 64.

Response: It was corrected in the revised manuscript.

Comment: Line 570, figure 32 and many other points afterward: what do “carbon H bond” and “covent H bond” exactly mean in the context of the ligand-receptor interactions? I never heard of them. I cannot find them in any of the cited references. Please, clarify better or tell me something more. Thank you.

Response: Nonclassical (weak) hydrogen bonds are formed between weak donors, such as C–H, and/or weak acceptors, such as π-electron density. They abound in organic compounds and biomolecules (proteins, carbohydrates) and are important contributors to their stability due to their large numbers. In the following the links to 5 articles describing the role of this type of bonds in chemistry and biological system.

https://pubs.acs.org/doi/10.1021/jacs.6b01249  https://pubs.rsc.org/en/content/articlehtml/2017/cp/c7cp02762a 

https://pubs.rsc.org/en/content/articlelanding/2013/ob/c3ob40828k

https://pubmed.ncbi.nlm.nih.gov/26751405/

https://www.eurjchem.com/index.php/eurjchem/article/view/1529

Comment: Scheme 7: change the “z” with the “O”.

Response: It was corrected in the revised manuscript.

Comment: Line 628: I cannot see the utility of HCl. Can you explain better?

Response: In the last step in scheme 7, HCl was used to convert icotinib to its hydrochloride salt. Scheme 7 was corrected to illustrate this step.

Comment: Line 648: Among must be written in capital initial letter.

Response: It was corrected in the revised manuscript.

Comment: Line 695: as for 64.

Response: It was corrected in the revised manuscript.

Comment: Line 702: use the present simple, not the present continuous.

Response: It was corrected in the revised manuscript.

Comment: Line 705: write “...revealed a growth…

Response: It was corrected in the revised manuscript.

Comment: Figure 42: the structure M7 is uncorrect. There is a CH2 more.

Response: We have revised the chemical structure of M7 reference 111. We found that the chemical structure provided in the manuscript is the same as that reported. Also, M7 has a molecular weight of 595 da, which exactly the same as the reported in ref 111. The chemical structure of M7 is provided as smile below

CS(=O)(C/C=[N]([O])\CC1=CC=C(C2=CC3=C(N=CN=C3NC4=CC(Cl)=C(OCC5=CC(F)=CC=C5)C=C4)C=C2)O1)=O

Comment: Line 721 and 722: geometric isomers and stereoisomeric isomers (used for the oximes) are exactly the same. Please use a uniform notation.

Response: It was corrected to “geometric isomers” in the revised manuscript.

Comment: Line 727-728: there no reference.

Response: Two references “[112,113] were added to this sentence

Comment: Scheme 9: compound 65 is a nitro-derivative that is reduced by Fe and acid. It is not an amino-derivative (see ref. 117). Further, the structure of 66 must be checked. I guess it is uncorrect.

We have revised ref 117 and we found compound 65 is not a nitro derivative. The article includes 4 schemes. The method reported by Gu, et al. was illustrated in the fourth scheme (No 4). Scheme 4 illustrates the method developed by the authors to prepare neratinib using Wittig–Horner reaction.

The chemical structure of comp. 66 is actually incorrect in the original reference (ref 117). It must be 2-(chloromethyl)pyridine to react with the phenolic OH of 65. Accordingly, we have corrected scheme 9 in the revised manuscript.

Comment: Line 764: the sentence lacks the verb!

Response: It was corrected in the revised manuscript.

Comment: Line 781: Also here lacks the verb.

Response: It was corrected in the revised manuscript.

Comment: Line 810: write “…treatment of patients…”

Response: It was corrected in the revised manuscript.

Comment: Line 831: “also” is not correctly placed.

Response: It was corrected in the revised manuscript.

Comment: Scheme 11: please, check the structures of compounds 82 and 83.

Response: The chemical structures of the 2 compounds were revised and corrected in the revised manuscript.

Line 887: what does “cellular potency” exactly mean?

Response: This sentence was revised and corrected in the revised manuscript.

Comment: Line 899: as for 64.

Response: It was corrected in the revised manuscript.

Comment: Line 905: as for 64.

Response: It was corrected in the revised manuscript.

Comment: Line 935: write “feces”, NOT “faces”

Response: It was corrected in the revised manuscript.

Comment: Line 955 write “were” (plural) instead of “was” (singular).

Response: It was corrected in the revised manuscript.

Comment: Line 986: “healthy males” of which animal?

Response: It was corrected in the revised manuscript.

Comment: Line 999: It is obvious that a “EGFR-TKI” acts by inhibition of a kinase

Response: This sentence was corrected in the revised manuscript.

Line 1042: write better the sentence. Maybe: “a half-life consistent with that of …”.

Response: This sentence was corrected

Comment: Line 1051-1052: the sentence is redundant. Read it carefully!

Response: The syntax of the text was revised and corrected in the revised manuscript.

Comment: Scheme 13: What is DMA? It is not clear.

Response: this acronym (DMF-DMA) was defined as dimethylformamide-dimethylacetal) in the manuscript.

Comment: Line 1059: Is 96 a “nitro group”? maybe a nitro compound.

Response: It was corrected to “reduction of the nitro group in 96”

Comment: Line 1064: I suppose it is not 88, but 99.

Response: It was corrected to “99” in the revised manuscript.

Comment: Line 1070: Why the word “mechanistic”

Response: It was corrected to “Investigation of the mechanism of action of …. “

Comment: Line 1095: the cell line MDA-MB-231 is repeated three times!!

Response: It was corrected in the revised manuscript. However, vandetanib was tested against MDA-MB-231 on the cellular level and in a xenograft model. So, this cell line was used 2 times in this paragraph.  

Comment: Line 1100: change “studies” with “studied”.

Response: It was replaced by “studies by” in the revised manuscript.

Comment: Line 1127: use the present perfect, NOT the past simple.

Response: It was corrected in the revised manuscript.

Comment: Line 1142: change “are being” with “have been”.

Response: It was corrected in the revised manuscript.

Comment: Line 1143-1146: is a repetition. Please delete it.

Response: It was deleted from the revised manuscript.

Comment: Line 1160: change “are being” with “have been”.

Response: It was corrected in the revised manuscript.

Comment: In conclusion, it is my opinion that the manuscript need to be partially rewritten and then carefully checked by reading it several times before to submit it further.

Response: Many parts of the manuscript were rewritten (indicated with a yellow highlighter in the revised manuscript). The manuscript was also revised 3 times before the submission